# Approximately Equivariant Neural Processes

**Matthew Ashman**[*]
University of Cambridge
mca39@cam.ac.uk

**Cristiana Diaconu**[*]
University of Cambridge
cdd43@cam.ac.uk

**Adrian Weller**
University of Cambridge
The Alan Turing Institute
aw665@cam.ac.uk

**Wessel Bruinsma**
Microsoft Research AI for Science
wessel.p.bruinsma@gmail.com

**Richard E. Turner**
University of Cambridge
Microsoft Research AI for Science
The Alan Turing Institute
ret23@cam.ac.uk

## Abstract

Equivariant deep learning architectures exploit symmetries in learning problems to improve the sample efficiency of neural-network-based models and their ability to generalise. However, when modelling real-world data, learning problems are often not *exactly* equivariant, but only approximately. For example, when estimating the global temperature field from weather station observations, local topographical features like mountains break translation equivariance. In these scenarios, it is desirable to construct architectures that can flexibly depart from exact equivariance in a data-driven way. Current approaches to achieving this cannot usually be applied out-of-the-box to any architecture and symmetry group. In this paper, we develop a general approach to achieving this using existing equivariant architectures. Our approach is agnostic to both the choice of symmetry group and model architecture, making it widely applicable. We consider the use of approximately equivariant architectures in neural processes (NPs), a popular family of meta-learning models. We demonstrate the effectiveness of our approach on a number of synthetic and real-world regression experiments, showing that approximately equivariant NP models can outperform both their non-equivariant and strictly equivariant counterparts.

## 1 Introduction

The development of equivariant deep learning architectures has spearheaded many advancements in machine learning, including CNNs [LeCun et al., 1989], group equivariant CNNs [Cohen and Welling, 2016a, Finzi et al., 2020], transformers [Vaswani et al., 2017], DeepSets [Zaheer et al., 2017], and GNNs [Scarselli et al., 2008]. When appropriate, equivariances provide useful inductive biases that can drastically improve sample complexity [Mei et al., 2021] and generalisation capabilities [Elesedy and Zaidi, 2021, Bulusu et al., 2021, Petrache and Trivedi, 2023] by exploiting symmetries present in the data. Yet, real-world data are seldom strictly equivariant. As an example, consider modelling daily precipitation across space and time. Such data may be close to translation equivariant—and therefore translation equivariance serves as a useful inductive bias—however, it is clear that it is not strictly translation equivariant due to geographical and seasonal variations. More generally, whilst there are aspects of modelling problems which are universal and exhibit symmetries such as equivariance (e.g. atmospheric physics) there are often unknown local factors (e.g. precise local topography) which break these symmetries and which the models do not have access to. It is thus desirable for the

---

[*]Equal contribution.

38th Conference on Neural Information Processing Systems (NeurIPS 2024).

model to be able to depart from strict equivariance when necessary; that is, to develop *approximately equivariant* models.

A family of models that have benefited greatly from the use of equivariant deep learning architectures are neural processes [NPs; Garnelo et al., 2018a,b]. However, many of the problem domains that NPs are applied to exhibit only approximate equivariance. It is therefore desirable to build approximately equivariant NPs through the use of approximately equivariant deep learning architectures. Whilst there exist several approaches to constructing approximately equivariant architectures, they are limited to CNN- and MLP-based models [Wang et al., 2022a, van der Ouderaa et al., 2022, Finzi et al., 2021, Romero and Lohit, 2022]—which restricts their applicability to certain symmetry groups—and often require modifications to the loss function used to train the models [Finzi et al., 2021, Kim et al., 2023a]. As there exist NPs that utilise a variety of architectures, such as transformers and GNNs [Carr and Wingate, 2019, Nguyen and Grover, 2022, Kim et al., 2019, Feng et al., 2022], these approaches are not sufficient. We address this shortcoming through the development of an approach to constructing approximately equivariant models that is both architecture and symmetry-group agnostic. Importantly, our approach can be realised without modifying the core architecture of existing equivariant models, enabling use in a variety of equivariant NPs such as the convolutional conditional NP [ConvCNP; Gordon et al., 2019], the equivariant CNP [EquivCNP; Kawano et al., 2021], the steerable CNP [SteerCNP; Holderrieth et al., 2021], the translation equivariant transformer NP [TE-TNP; Ashman et al., 2024], and the relational CNP [RCNP; Huang et al., 2023].

We outline our core contributions as follows:

1. We demonstrate that under certain technical conditions, such as Hölder regularity and compactness, any non-equivariant mapping between function spaces can be approximated by an equivariant mapping with fixed additional inputs (Theorems 2 and 3). This provides insight into how to construct approximately equivariant models, and generalises several existing approaches to relaxing equivariant constraints.

2. We apply this result and the insights it provides to construct approximately equivariant versions of several popular equivariant NPs. The modifications required are *very simple, yet effective*: we demonstrate improved performance relative to both strictly equivariant and non-equivariant models on a number of spatio-temporal regression tasks.

## 2 Background

We consider the supervised learning setting, where $\mathcal{X}$, $\mathcal{Y}$ denote the input and output spaces, and $(\mathbf{x}, \mathbf{y}) \in \mathcal{X} \times \mathcal{Y}$ denotes an input-output pair. Let $\mathcal{S} = \bigcup_{N=0}^{\infty} (\mathcal{X} \times \mathcal{Y})^N$ be a collection of all finite data sets, which includes the empty set $\varnothing$. Let $\mathcal{S}_M = (\mathcal{X} \times \mathcal{Y})^M$ be the collection of $M$ input-output pairs and $\mathcal{S}_{\leq M} = \bigcup_{m=1}^{M} \mathcal{S}_m$ be the collection of at most $M$ pairs. We denote a context and target set with $\mathcal{D}_c, \mathcal{D}_t \in \mathcal{S}$, where $|\mathcal{D}_c| = N_c$, $|\mathcal{D}_t| = N_t$. Let $\mathbf{X}_c \in (\mathcal{X})^{N_c}$, $\mathbf{Y}_c \in (\mathcal{Y})^{N_c}$ be the inputs and corresponding outputs of $\mathcal{D}_c$, and let $\mathbf{X}_t \in (\mathcal{X})^{N_t}$, $\mathbf{Y}_t \in (\mathcal{Y})^{N_t}$ be defined analogously. We denote a single task as $\xi = (\mathcal{D}_c, \mathcal{D}_t) = ((\mathbf{X}_c, \mathbf{Y}_c), (\mathbf{X}_t, \mathbf{Y}_t))$. Let $\mathcal{P}(\mathcal{X})$ denote the collection of $\mathcal{Y}$-valued stochastic processes on $\mathcal{X}$. Let $\Theta$ denote the parameter space for some family of probability densities, e.g. means and variances for the space of all Gaussian densities.

### 2.1 Neural Processes

NPs [Garnelo et al., 2018a,b] can be viewed as neural-network-based parametrisations of *prediction maps* $\pi \colon \mathcal{S} \to \mathcal{P}(\mathcal{X})$ from data sets $\mathcal{S}$ to predictions $\mathcal{P}(\mathcal{X})$, where predictions are represented by $\mathcal{Y}$-valued stochastic processes on $\mathcal{X}$ [Foong et al., 2020, Bruinsma, 2022]. Throughout, we denote the density of the finite-dimensional distribution of $\pi(\mathcal{D})$ at inputs $\mathbf{X}$ by $p(\cdot \mid \mathbf{X}, \mathcal{D})$. In this work, we restrict our attention to conditional NPs [CNPs; Garnelo et al., 2018a], which only target marginal predictive distributions by assuming that the predictive densities factorise: $p(\mathbf{Y}|\mathbf{X}, \mathcal{D}) = \prod_n p(\mathbf{y}_n|\mathbf{x}_n, \mathcal{D})$. We denote all parameters of a CNP by $\omega$. CNPs are trained in a meta-learning fashion, in which the expected predictive log-probability is maximised:

$$\omega_{\mathrm{ML}} = \arg\max_{\omega} \mathcal{L}_{\mathrm{ML}}(\omega) \quad \text{where} \quad \mathcal{L}_{\mathrm{ML}}(\omega) = \mathbb{E}_{p(\xi)}\Big[ \sum_{n=1}^{N_t} \log p_{\omega}(\mathbf{y}_{t,n}|\mathbf{x}_{t,n}, \mathcal{D}_c) \Big]. \quad (1)$$

Here, the expectation is taken with respect to the distribution over tasks $p(\xi)$. In practice, we only have access to a finite number of tasks for training, so the expectation is approximated with an average

over tasks. The global maximum is achieved if and only if the model recovers the ground-truth marginals [Proposition 3.26 by Bruinsma, 2022].

## 2.2 Group Equivariance

We consider equivariance with respect to transformations in some group $G$. For example, $G$ can be the group of translations or the group of rotations. Mathematically, a group $G$ is a set endowed with a binary operation $G \times G \to G$ (denoted as multiplication) such that (i) $(fg)h = f(gh)$ for all $f, g, h \in G$; (ii) there exists an identity element $e \in G$ such that $eg = ge = g$ for all $g \in G$; and (iii) every element $g \in G$ has an inverse $g^{-1} \in G$. A $G$-space is a space $X$ for which there exists a function $G \times X \to X$ called a group action (again denoted by multiplication) such that (i) $ex = x$ for all $x \in X$; and (ii) $f(gx) = (fg)x$ for all $f, g \in G$ and $x \in X$. The notion of group equivariance is used to describe mappings for which, when the input to the mapping is transformed by some $g \in G$, the output of the mapping is transformed equivalently. This is formalised in the following definition.

**Definition 1** ($G$-equivariance)**.** *Let $X$ and $Y$ be $G$-spaces. Call a mapping $\rho \colon X \to Y$ $G$-equivariant if $\rho(gx) = g\rho(x)$ for all $g \in G$ and $x \in X$.*

## 2.3 Group-Equivariant Conditional Neural Processes

The property of $G$-equivariance is particularly useful in NPs when modelling ground-truth stochastic processes that exhibit $G$-stationarity:

**Definition 2** ($G$-stationary stochastic process)**.** *Let $\mathcal{X}$ be a $G$-space. We say that a stochastic process $P \in \mathcal{P}(\mathcal{X})$ is $G$-stationary if, for $f \sim P$ and all $g \in G$, $f(\cdot)$ is equal in distribution to $f(g \cdot)$.*

Let $P \in \mathcal{P}(\mathcal{X})$ be a ground-truth stochastic process. For a dataset $\mathcal{D} \in \mathcal{S}$, define $\pi_P(\mathcal{D})$ by integrating $P$ against some density $\pi'_P(\mathcal{D})$, such that $\mathrm{d}\pi_P(\mathcal{D}) = \pi'_P(\mathcal{D}) \, \mathrm{d}P$. $\pi'_P(\mathcal{D})$ is the Radon-Nikodym derivative of the posterior stochastic process with respect to the prior; intuitively, $\pi'_P(\mathcal{D})(f)$ is proportional to the likelihood, $\pi'_P(\mathcal{D})(f) = p(\mathcal{D} \mid f)/p(\mathcal{D})$, so it determines how the data is observed (e.g. under which noise?). Assume that $\pi'_P(\varnothing) \propto 1$ so that $\pi_P(\varnothing) = P$. We say that $\pi'_P$ is $G$-invariant if $\pi'_P(g\mathcal{D}) \circ g = \pi'_P(\mathcal{D})$.

**Proposition 1** ($G$-stationarity and $G$-equivariance)**.** *The ground-truth stochastic process $P$ is $G$-stationary and $\pi'_P$ is $G$-invariant if, and only if, $\pi_P$ is $G$-equivariant.*

*Proof.* This is Thm 2.1 by Ashman et al. [2024] with translations replaced by applications of $g \in G$. $\qquad\square$

Thus, for data generated from a stochastic process that is approximately $G$-stationary, incorporating $G$-equivariance into NP approximations of the corresponding prediction map can serve as a useful inductive bias that can help generalisation and improve parameter efficiency. Intuitively, requiring a NP to be equivariant effectively ties parameters together, which significantly reduces the search space during optimisation, enabling better solutions to be found with fewer data. The improved generalisation capabilities of $G$-equivariant models is formalised by Elesedy and Zaidi [2021], Bulusu et al. [2021], Petrache and Trivedi [2023].

For a CNP, assume that every marginal $p(\cdot \mid \mathbf{x}, \mathcal{D})$ is in some fixed parametric family with parameters $\theta(\mathbf{x}; \mathcal{D}) \in \Theta$. For example, $\theta$ could consist of the mean and variance for a Gaussian distribution. Let $C(\mathcal{X}, \Theta)$ denote the set of continuous functions $\mathcal{X} \to \Theta$. For a CNP, define the associated *parameter map* $\Phi \colon \mathcal{S} \to C(\mathcal{X}, \Theta)$ by $\Phi(\mathcal{D})(\mathbf{x}) = \theta(\mathbf{x}; \mathcal{D})$. Intuitively, the parameter map $\Phi$ maps a dataset $\mathcal{D}$ to a function $\Phi(\mathcal{D})$ giving the parameters $\Phi(\mathcal{D})(\mathbf{x})$ for every input $\mathbf{x}$. Assume that $\mathcal{X}$ is a $G$-space. We turn $\mathcal{S}$ and $C(\mathcal{X}, \Theta)$ into $G$-spaces by applying $g$ to the inputs: $g\mathcal{D} \in \mathcal{S}$ consists of the input–output pairs $(g\mathbf{x}_n, \mathbf{y}_n)$, and $g\theta(\mathbf{x}; \mathcal{D}) \in C(\mathcal{X}, \Theta)$ is defined by $g\theta(\mathbf{x}; \mathcal{D}) = \theta(g^{-1}\mathbf{x}; \mathcal{D})$.

**Definition 3** ($G$-equivariant CNP)**.** *A CNP is $G$-equivariant if the associated parameter map $\Phi$ is $G$-equivariant: $\Phi(g\mathcal{D}) = g\Phi(\mathcal{D})$ for all datasets $\mathcal{D}$.*

To parametrise a $G$-equivariant CNP, we must parametrise the associated parameter map $\Phi$. Here we present a general construction by Kawano et al. [2021].

**Theorem 1** (Representation of $G$-equivariant CNPs, Theorem 2 by Kawano et al. [2021])**.** *Let $\mathcal{Y} \subseteq \mathbb{R}^D$ be compact. Consider an appropriate collection $\mathcal{S}'_{\leq M} \subseteq \mathcal{S}_{\leq M}$. Then a function $\Phi \colon \mathcal{S}'_{\leq M} \to$*

$C(\mathcal{X}, \Theta)$ *is continuous, permutation-invariant, and $G$-equivariant if and only if it is of the form*

$$\Phi(\mathcal{D}) = \rho(f_{enc}(\mathcal{D})) \quad where \quad f_{enc}((\mathbf{x}_1, \mathbf{y}_1), \dots, (\mathbf{x}_m, \mathbf{y}_m)) = \sum_{i=1}^m \phi(\mathbf{y}_i) \psi(\,\cdot\,, \mathbf{x}_i) \qquad (2)$$

*for some continuous $\phi \colon \mathcal{Y} \to \mathbb{R}^{2D}$, an appropriate $G$-invariant positive-definite kernel $\psi \colon \mathcal{X}^2 \to \mathbb{R}$ (i.e. $\psi(\mathbf{x}_n, \mathbf{x}_m) = \psi(g\mathbf{x}_n, g\mathbf{x}_m)$ for all $g \in G$), and some continuous and $G$-equivariant $\rho \colon \mathbb{H} \to C(\mathcal{X}, \Theta)$ with $\mathbb{H}$ an appropriate $G$-invariant space of functions (i.e. closed under $g \in G$).*

Theorem 1 naturally gives rise to architectures which can be deconstructed into two components: an *encoder* and a *decoder*. The encoder, $f_{enc} \colon \mathcal{S} \to \mathcal{Z}$, maps datasets to some embedding space $\mathcal{Z}$. The decoder, $\rho \colon \mathcal{Z} \to C(\mathcal{X}, \Theta)$, takes this representation and maps to a function that gives the parameters of the CNP's predictive distributions: $p(\mathbf{y}|\mathbf{x}, \mathcal{D}) = p(\mathbf{y}|\rho(f_{enc}(\mathcal{D}))(\mathbf{x}))$ where $\rho(f_{enc}(\mathcal{D}))(\mathbf{x}) \in \Theta$. For simplicity, we often view the decoder as a function $\mathcal{Z} \times \mathcal{X} \to \Theta$ and more simply write $\rho(f_{enc}(\mathcal{D}), \mathbf{x})$. In Theorem 1, both the encoder $f_{enc}$ and decoder $\rho$ are $G$-equivariant. Many neural process architectures are of this form, including the ConvCNP [Gordon et al., 2019], the EquivCNP [Kawano et al., 2021], the RCNP [Huang et al., 2023], and the TE-TNP [Ashman et al., 2024].[2] We discuss the specifics of each of these architectures in Appendix B.

## 3 Equivariant Decomposition of Non-Equivariant Functions

In this section, we demonstrate that, subject to regularity conditions, any *non-equivariant* operator between function spaces can be constructed as, or approximated by, an *equivariant* mapping with additional, fixed functions as input. These results motivate a simple construction for approximately equivariant models, which we use to construct approximately equivariant NPs in Section 3.1. We first illustrate the proof technique by proving the result for linear operators (Theorem 2) and then extend the result to nonlinear operators (Theorem 3).

**Setup.** Let $(\mathbb{H}, \langle \,\cdot\,, \,\cdot\, \rangle)$ be a Hilbert space of functions $\mathcal{X} \to \mathbb{R}$. Let $G$ be a group acting linearly on $\mathbb{H}$ from the left. For every $g \in G$ and $f \in \mathbb{H}$, applying $g$ to $f$ gives another function: $gf \in \mathbb{H}$. By linearity, for $f_1, f_2 \in \mathbb{H}$, $g(f_1 + f_2) = gf_1 + gf_2$. Assume that $\mathbb{H}$ is separable and that $\mathbb{H}$ is $G$-invariant, meaning that, for all $f_1, f_2 \in \mathbb{H}$ and $g \in G$, $\langle gf_1, gf_2 \rangle = \langle f_1, f_2 \rangle$. If $\mathbb{H} = L^2(\mathcal{X})$ and $\mathcal{X}$ is a separable metric space, then $\mathbb{H}$ is also separable [Brezis, 2011, Theorem 4.13]. Let $B$ be the collection of bounded linear operators on $\mathbb{H}$. We say that an operator $T \in B$ is *of finite rank* if the dimensionality of the range of $T$ is finite. Moreover, we say that an operator $T \in B$ is *compact* if the image of every bounded set is relatively compact. Let $(e_i)_{i \geq 1}$ be an orthonormal basis for $\mathbb{H}$. Define $P_n = \sum_{i=1}^n e_i \langle e_i, \,\cdot\, \rangle$. The following proposition is well known:

**Proposition 2** (Finite-rank approximation of compact operators; e.g., Corollary 6.2 by Brezis [2011].)**.** *Let $T \in B$. Then $T$ is compact if and only if there exists a sequence of operators of finite rank $(T_n)_{n \geq 1} \subseteq B$ such that $\|T - T_n\| \to 0$. In particular, one may take $T_n = P_n T$, so $T$ is a compact operator if and only if $\|T - P_n T\| \to 0$.*

Intuitively, compactness of an operator implies that its inputs and outputs can be well approximated with a finite-dimensional basis. The following new result shows that every compact $T \in B$ can be approximated with a $G$-equivariant function with additional fixed inputs: $T \approx E_n(\,\cdot\,, t_1, \dots, t_{2n})$ where $E_n$ is $G$-equivariant and $t_1, \dots, t_{2n}$ are the additional fixed inputs.

**Theorem 2** (Approximation of non-equivariant linear operators.)**.** *Let $T \in B$. Assume that $T$ is compact. Then there exists a sequence of continuous nonlinear operators $E_n \colon \mathbb{H}^{1+2n} \to \mathbb{H}$, $n \geq 1$, and a sequence of functions $(t_n)_{n \geq 1} \subseteq \mathbb{H}$ such that every $E_n$ is $G$-equivariant,*

$$E_n(gf_1, gf_2, \dots, gf_{2n+1}) = gE_n(f_1, f_2, \dots, f_{2n+1}) \quad for\ all\ g \in G\ and\ f_1, \dots, f_{2n+1} \in \mathbb{H}, \quad (3)$$

*and $\|T - E_n(\,\cdot\,, t_1, \dots, t_{2n})\| \to 0$.*

*Proof.* Observe that $P_n T f = \sum_{i=1}^n e_i \langle T^* e_i, f \rangle = \sum_{i=1}^n e_i \langle \tau_i, f \rangle$, with $\tau_i = T^* e_i \in \mathbb{H}$. Define the continuous nonlinear operators $E_n \colon \mathbb{H}^{1+2n} \to \mathbb{H}$ as $E_n(f, \tau_1, e_1, \dots, \tau_n, e_n) = \sum_{i=1}^n e_i \langle \tau_i, f \rangle$. By $G$-invariance of $\mathbb{H}$, these operators are $G$-equivariant:

$$E_n(gf, g\tau_1, ge_1, \dots) = \sum_{i=1}^n ge_i \langle g\tau_i, gf \rangle = g \sum_{i=1}^n e_i \langle \tau_i, f \rangle = gE_n(f, \tau_1, e_1, \dots). \qquad (4)$$

---

[2]We note that the form of the embedded dataset $f_{enc}(\mathcal{D})$ differs slightly in the RCP and TE-TNP. However, in both cases the form in Equation 2 can be recovered as special cases. We discuss this more in Appendix B.

Since $T$ is compact, $\|T - P_n T\| \to 0$ (Proposition 2), so $\|T - E_n(\,\cdot\,, \tau_1, \ldots, \tau_n, e_1, \ldots, e_n)\| \to 0$, which proves the result. $\qquad\square$

As an example, consider $G$ to be the group of translations such that $E_n$ is translation equivariant. Translation-equivariant mappings between function spaces can be approximated with a CNN [Kumagai and Sannai, 2020]. Therefore, Theorem 2 gives that $T \approx \mathrm{CNN}(\,\cdot\,, t_1, \ldots, t_k)$ where $t_1, \ldots, t_k$ are additional fixed inputs that are given as additional channels to the CNN and can be treated as model parameters to be optimised. These additional inputs in the CNN break translation equivariance, because they are not translated whenever the original input is translated. The number $k$ of such inputs roughly determines to what extent equivariance is broken: the larger $k$, the more non-equivariant the approximation becomes. This example holds true for any CNN architecture provided it can approximate any translation-equivariant operator.

Theorem 2 is not exactly what we set out to achieve: we wish to construct approximately equivariant mappings, not use equivariant mappings to approximate non-equivariant models. However, instead of applying Theorem 2 directly to $T$, consider the decomposition $T = T_{\mathrm{equiv}} + T_{\mathrm{non\text{-}equiv}}$ where $T_{\mathrm{equiv}}$ is in some sense the best "equivariant approximation" of $T$ and $T_{\mathrm{non\text{-}equiv}}$ is the residual. Then, if we approximate $T_{\mathrm{non\text{-}equiv}}$ with $E_{\mathrm{non\text{-}equiv}}$ using Theorem 2, and approximate $T_{\mathrm{equiv}}$ with some $G$-equivariant mapping $E_{\mathrm{equiv}}$ directly, we have $T \approx E_{\mathrm{equiv}} + E_{\mathrm{non\text{-}equiv}}(\,\cdot\,, t_1, \ldots, t_k)$. If $k = 0$, this approximation roughly recovers $E_{\mathrm{equiv}}$, the best equivariant approximation of $T$; and, if $k$ is increased, the equivariance starts to break and the approximation starts to wholly approximate $T$. Specifically, in Theorem 2, $k = 2n$ determines the dimensionality of the range of the approximation $E_n$. Therefore, a small $k$ means that that one would deviate from $T_{\mathrm{equiv}}$ in only a few degrees of freedom. The idea that $k$ can be used to control the degree to which our approximation is $G$-equivariant inspires our training procedure in Section 3.1, where, with some probability, we set the additional inputs to zero. This forces the model to learn the 'best equivariant approximation', so that the number of additional fixed inputs has the desired influence of controlling only the flexibility of the non-equivariant component.

We now extend the result to any continuous, possibly nonlinear operator $T\colon \mathbb{H} \to \mathbb{H}$. For $T \in B$, it is true that $T$ is compact if and only if $\|T - P_n T P_n\| \to 0$ (Proposition 3 in Appendix A). In generalising Theorem 2 to nonlinear operators, we shall use this equivalent condition. Roughly speaking, $\|T - P_n T P_n\| \to 0$ says that both the domain and range of $T$ admit a finite-dimensional approximation, and the proof then proceeds by discretising these finite-dimensional approximations.

**Theorem 3** (Approximation of non-equivariant operators.). *Let $T\colon \mathbb{H} \to \mathbb{H}$ be a continuous, possibly nonlinear operator. Assume that $\|T - P_n T P_n\| \to 0$, and that $T$ is $(c, \alpha)$-Hölder for $c, \alpha > 0$, in the following sense:*

$$\|T(u) - T(v)\| \le c\|u - v\|^{\alpha} \quad \text{for all } u, v \in \mathbb{H}. \tag{5}$$

*Moreover, assume that the orthonormal basis $(e_i)_{i \ge 1}$ is chosen such that, for every $n \in \mathbb{N}$ and $g \in G$, $\operatorname{span}\{e_1, \ldots, e_n\} = \operatorname{span}\{ge_1, \ldots, ge_n\}$, meaning that subspaces spanned by finitely many basis elements are invariant under the group action $(*)$. Let $M > 0$. Then there exists a sequence $(k_n)_{n \ge 1} \subseteq \mathbb{N}$, a sequence of continuous nonlinear operators $E_n\colon \mathbb{H}^{1+k_n} \to \mathbb{H}$, $n \ge 1$, and a sequence of functions $(t_n)_{n \ge 1} \subseteq \mathbb{H}$ such that every $E_n$ is $G$-equivariant,*

$$E_n(gf_1, \ldots, gf_{1+k_n}) = gE_n(f_1, \ldots, f_{1+k_n}) \quad \text{for all } g \in G \text{ and } f_1, \ldots, f_{1+k_n} \in \mathbb{H}, \tag{6}$$

*and*

$$\sup_{u \in \mathbb{H}: \|u\| \le M} \|T(u) - E_n(u, t_1, \ldots, t_{k_n})\| \to 0. \tag{7}$$

*If assumption $(*)$ does not hold, then the conclusion holds with $E_n(\,\cdot\,, t_1, \ldots, t_{k_n})$ replaced by $E_n(P_n\,\cdot\,, t_1, \ldots, t_{k_n})$. We provide a proof in Appendix A.*

Condition $(*)$ says that the orthonormal basis $(e_i)_{i \ge 1}$ must be chosen in a way such that applying the group action does not "introduce higher basis elements". Note that only one such basis needs to exist for the result to hold. An important example where this condition holds is $\mathbb{H} = L^2(\mathbb{S}^1)$ where $\mathbb{S}^1 = \mathbb{R}/\mathbb{Z}$ is the one-dimensional torus ($[0, 1]$ with endpoints identified); $G$ the one-dimensional translation group; and $(e_i)_{i \ge 1}$ the Fourier basis: $e_k(x) = e^{i2\pi k x}$. Then $e_k(x - \tau) = e^{i2\pi k \tau} e_k(x) \propto e_k(x)$, so $\operatorname{span}\{e_k(\,\cdot\, - \tau)\} = \operatorname{span}\{e_k\}$. This example shows that the result holds for translation equivariance, which is the symmetry that we will primarily consider in the experiments. Importantly, Theorem 3 requires $\|T - P_n T P_n\| \to 0$ to hold. For this example, this condition roughly means that the dependence of $T$ on high-frequency basis elements goes to zero as the frequency increases.

In Theorem 3, the sequence $(k_n)_{n \geq 1}$ determines how many additional fixed inputs are required to obtain a good approximation. For linear $T$ (Theorem 2), $k_n$ grows linearly in $n$. In the nonlinear case, $k_n$ may grow faster than linear in $n$, so more basis functions may be required. We leave it to future work to more accurately identify how the growth of $k_n$ in $n$ depends on $T$. A promising idea is to consider $\|Tgf - gTf\|$ as a measure of how equivariant a mapping is [Wang et al., 2022a].

## 3.1 Approximately Equivariant Neural Processes

Theorem 3 provides a general construction of non-equivariant functions from equivariant functions, lending insight into how approximately equivariant neural processes can be constructed. Consider a $G$-equivariant CNP with $G$-equivariant encoder $f_{\text{enc}}$ and decoder $\rho$. The construction that we consider in this paper is to insert additional fixed inputs into the decoder $\rho$:

$$p(\mathbf{Y}|\mathbf{X}, \mathcal{D}) = \prod_n p(\mathbf{y}_n|\mathbf{x}_n, \mathcal{D}) = \prod_n p(\mathbf{y}_n|\mathbf{x}_n, \rho(f_{\text{enc}}(\mathcal{D}), t_1, \ldots, t_B)) \tag{8}$$

where the decoder $\rho \colon \mathbb{H}^{1+B} \to C(\mathcal{X}, \Theta)$ now takes in the dataset embedding $f_{\text{enc}}(\mathcal{D})$ as well as $B$ additional fixed inputs $(t_b)_{b=1}^{B} \subseteq \mathbb{H}$. These become additional model parameters and break $G$-equivariance of the decoder. The more are included, the more non-equivariant the decoder becomes. Crucially, Theorem 3 shows that including sufficiently many additional inputs eventually recovers any non-equivariant decoder. Conversely, by only including a few of them, the decoder deviates from $G$-equivariance in only a few degrees of freedom. Hence, the number $B$ of additional fixed inputs determines to which extent the decoder can become non-equivariant.

There are a myriad of ways in which $(t_b)_{b=1}^{B}$ can be incorporated into existing architectures for $\rho$. If $\rho$ is a CNN or a $G$-equivariant CNN, the additional inputs become additional channels, which can be either concatenated or added to the dataset embedding. We use the latter approach in our implementation. For more general $\rho$, such as that used in the TE-TNP, we can employ effective alternative choices, as discussed in Appendix B. Theorem 3 also intimately connects to positional embeddings in transformer-based LLMs [Vaswani et al., 2017] and vision transformers [ViT; Dosovitskiy et al., 2021]. These architectures consist of stacked multi-head self-attention (MHSA) layers, which are permutation equivariant with respect to the input tokens. However, after tokenising individual words or pixel values into tokens, the underlying transformer which processes these tokens should not be equivariant with respect to the permutations of words or pixels, as their position is important to the overall structure. Following Theorem 3, we can add word- or pixel-position-specific inputs to break this equivariance. This corresponds exactly to the usual positional embeddings that are regularly used. This connection between breaking equivariance and positional encodings was also noted by Lim et al. [2023], where they interpret several popular positional encodings as group representations that can help incorporate approximate equivariance.

**Recovering equivariance out-of-distribution.** We stress that equivariance is crucial for models to generalise beyond the training distribution—this is the 'shared' component that is inherent to the system we are modelling. Whilst the non-equivariant component is able to learn local symmetry-breaking features that are revealed with sufficient data, these features do not reveal themselves outside the training domain. To obtain optimal generalisation performance, we desire that the model ignores the non-equivariant component outside of the training domain and instead reverts to equivariant predictions. We achieve this in the following way: (i) During training, set $t_b = 0$ entirely with some fixed probability. This allows the model to learn and produce predictions for the equivariant component of the underlying system wherever the fixed additional inputs are zero. (ii) Forcefully set the additional fixed inputs to zero outside the training domain: $t_b(\mathbf{x}) = 0$ for $\mathbf{x}$ not close to any training data.[3] This forces the model to revert to equivariant predictions outside the training domain, which should substantially improve the model's ability to generalise.

Our approach to recovering equivariance out-of-distribution also connects to Wang et al.'s (2022b) notion of $\epsilon$-approximate equivariance. Let $\pi_{\text{AE-NP}}$ be the learned neural process, and let $\pi_{\text{E-NP}}$ be the neural process obtained by setting $t_b = 0$ in $\pi_{\text{AE-NP}}$. If the predictions of $\pi_{\text{AE-NP}}$ are not too different from those of $\pi_{\text{E-NP}}$, because the fixed additional inputs only affect the predictions in a limited way, then $\pi_{\text{AE-NP}}$ is $\epsilon$-approximately equivariant in the sense of Wang et al. [2022b]. See Appendix D.

---

[3]For example, the rectangle covering the maximum longitude and latitude of geographical locations seen during training of an environmental model.

# 4 Related Work

**Group equivariance and equivariant neural processes.** Group equivariant deep learning architectures are ubiquitous in machine learning, including CNNs [LeCun et al., 1989], group equivariant CNNs [Cohen and Welling, 2016a], transformers [Vaswani et al., 2017, Lee et al., 2019] and GNNs [Scarselli et al., 2008]. This is testament to the usefulness of incorporating inductive biases into models, with a significant body of research demonstrating improved sample complexity and generalisation capabilities [Mei et al., 2021, Elesedy and Zaidi, 2021, Bulusu et al., 2021, Zhu et al., 2021]. There exist a number of NP models which realise these benefits. Notably, the ConvCNP [Gordon et al., 2019], RCNP [Huang et al., 2023], and TE-TNP [Ashman et al., 2024] all build translation equivariance into NPs through different architecture choices. More general equivariances have been considered by both the EquivCNP [Kawano et al., 2021] and SteerCNPs [Holderrieth et al., 2021], both of which consider architectures similar to the ConvCNP.

**Approximate group equivariance.** Two methods similar to ours are those of Wang et al. [2022a] and van der Ouderaa et al. [2022], who develop approximately equivariant architectures for group equivariant and steerable CNNs [Cohen and Welling, 2016a,b]. We demonstrate in Appendix C that both approaches are specific examples of our more general approach. Finzi et al. [2021] and Kim et al. [2023a] obtain approximate equivariance through a modification to the loss function such that the equivariant subspace of linear layers is favoured. These methods are less flexible than our approach, which can be applied to *any equivariant architecture*. Romero and Lohit [2022] only enforce strict equivariance for specific elements of the symmetry group considered. Their approach hinges on the ability to construct and sample from probability distributions over elements of the groups, which both restricts their applicability and complicates their implementation. An orthogonal approach to imbuing models with approximate equivariance is through data augmentation. Data augmentation is trivial to implement; however, previous work [Wang et al., 2022b] has demonstrated that both equivariant and approximately equivariant models achieve better generalisation bounds than data augmentation. Kim et al. [2023b] propose a similar approach to data augmentation, replacing a uniform average over group transformations with an expectation over an input-dependent probability distribution. Implementation of this distribution is involved, and must be hand-crafted for each symmetry group.

# 5 Experiments

In this section, we evaluate the performance of a number of approximately equivariant NPs derived from existing strictly equivariant NPs in modelling both synthetic and real-world data. We provide detailed descriptions of the architectures and datasets used in Appendix E.[4] Throughout, we postfix to the name of each model the group it is equivariant with respect to, with the postfix $\widetilde{G}$ denoting approximate equivariance with respect to group $G$. We shall also omit reference to the dimension when denoting a symmetry group (e.g. $T(n)$ becomes $T$). In all experiments, we compare the performance of three TNP-based models [Ashman et al., 2024]: non-equivariant, $T$, and $\widetilde{T}$; three ConvCNP-based models [Gordon et al., 2019]: $T$, $\widetilde{T}$ using the approach described in Section 3.1, and $\widetilde{T}$ using the relaxed CNN approach of Wang et al. [2022a]; and two EquivCNP-based models [Kawano et al., 2021]: $E$ and $\widetilde{E}$. For experiments involving a large number of datapoints, we replace the TNP-based models with pseudo-token TNP-based models [PT-TNP; Ashman et al. 2024].

## 5.1 Synthetic 1-D Regression With the Gibbs Kernel

We begin with a synthetic 1-D regression task with datasets drawn from a Gaussian process (GP) with the Gibbs kernel [Gibbs, 1998]. The Gibbs kernel similar to the squared exponential kernel, except the lengthscale $\ell(x)$ is a function of position $x$. The non-stationarity of the kernel implies that the predictive map is not translation equivariant, hence we expect an improvement of approximately equivariant NPs with respect to their equivariant counterparts. We construct each dataset by first sampling a change point, either side of which the GP lengthscale is either small ($\ell(x) = 0.1$) or large ($\ell(x) = 4.0$). The range from which the context and target points are sampled is itself randomly sampled, so that the change point is not always present in the data. We sample $N_c \sim \mathcal{U}\{1, 64\}$ and the number of target points is set as $N_t = 128$. See Appendix E for a complete description.

---

[4]An implementation of our models can be found at cambridge-mlg/aenp.

**Table 1:** Average test log-likelihoods (↑) for the synthetic 1-D GP and 2-D smoke experiments. For the 1-D dataset we used the regular TNP, while for the 2-D experiment we used the PT-TNP. Results are grouped together by model class. Best in-class result is bolded.

| | 1-D GP | | 2-D Smoke |
|---|---|---|---|
| Model | ID Log-lik. (↑) | OOD Log-lik. (↑) | Log-lik. (↑) |
| TNP | $-\mathbf{0.406} \pm \mathbf{0.004}$ | $-1.3734 \pm 0.002$ | $4.299 \pm 0.008$ |
| TNP ($T$) | $-0.500 \pm 0.004$ | $-\mathbf{0.430} \pm \mathbf{0.007}$ | $4.181 \pm 0.011$ |
| TNP ($\widetilde{T}$) | $-\mathbf{0.406} \pm \mathbf{0.004}$ | $-0.424 \pm 0.007$ | $\mathbf{4.715} \pm \mathbf{0.010}$ |
| ConvCNP ($T$) | $-0.499 \pm 0.004$ | $-0.442 \pm 0.006$ | $3.637 \pm 0.041$ |
| ConvCNP ($\widetilde{T}$) | $-0.430 \pm 0.004$ | $-\mathbf{0.412} \pm \mathbf{0.006}$ | $3.827 \pm 0.011$ |
| RelaxedConvCNP ($\widetilde{T}$) | $-\mathbf{0.419} \pm \mathbf{0.004}$ | $-0.405 \pm 0.007$ | $\mathbf{4.006} \pm \mathbf{0.010}$ |
| EquivCNP ($E$) | $-0.504 \pm 0.004$ | $-0.443 \pm 0.007$ | $4.194 \pm 0.015$ |
| EquivCNP ($\widetilde{E}$) | $-\mathbf{0.435} \pm \mathbf{0.004}$ | $-\mathbf{0.413} \pm \mathbf{0.007}$ | $\mathbf{4.233} \pm \mathbf{0.012}$ |

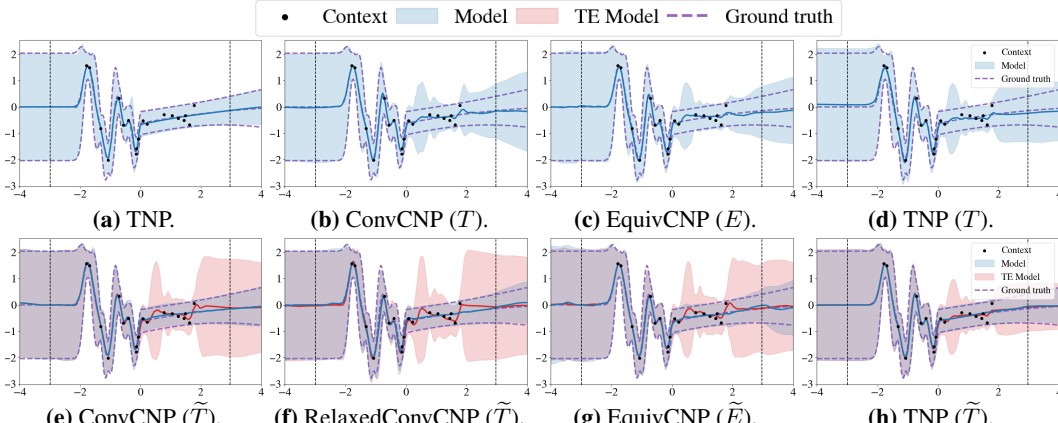

**Figure 1:** A comparison between the predictive distributions on a single synthetic 1-D regression dataset of the TNP-, ConvCNP-, and EquivCNP-based models. For the approximately equivariant models, we plot both the model's predictive distribution (blue), as well as the predictive distributions obtained without using the fixed inputs (red). The dotted black lines indicate the target range.

We evaluate the log-likelihood on both the in-distribution (ID) training domain and on an out-of-distribution (OOD) setting in which the test domain is far away from the change point. Table 1 presents the results. We observe that the approximate equivariant models are able to: 1) recover the performance of the non-equivariant TNP within the non-equivariant ID regime; and 2) generalise as well as the equivariant models when tested OOD. We provide an illustrative comparison of the predictive distributions for each model in Figure 1. More examples can be found in Appendix E. When transitioning from the low-lengthscale to the high-lengthscale region, the equivariant predictive distributions behave as though they are in the low-lengthscale region. This is due to the ambiguity as to whether the high-lengthscale region has been entered. In contrast, the approximately equivariant models are able to learn that a change point always exists at $x = 0$, resolving this ambiguity.

We show in Appendix E.1 that the approximately equivariant models deviate only slightly from strict equivariance, thus being equivariant in the approximate sense as we describe in Section 3.1. We also analyse the performance of the models with increasing number of basis functions in Appendix E.1. The biggest improvement is obtained when going from 0 to 1 basis function, highlighting the importance of relaxing strict equivariance. We generally find that a few basis functions suffice to capture the symmetry-breaking features, but using more fixed inputs than necessary does not hurt performance, thus justifying in practice the use of a 'sufficiently high number' of fixed inputs.

## 5.2 Smoke Plumes

There are inherent connections between symmetries and dynamical systems, yet it is also true that real world dynamics rarely exhibit perfect symmetry. Motivated by this, we investigate the utility of

approximate equivariance in the context of modelling symmetrically-imperfect simulations generated from partial differential equations. We consider a setup similar to Wang et al. [2022a]—we construct a dataset of $128 \times 128$ 2-D smoke simulations, computing the air flow in a closed box with a smoke source. Besides the closed boundary, we also introduce a fixed spherical obstacle through which smoke cannot pass, and we sample the position of the spherical smoke inflow out of three possible locations. These three components break the symmetry of the system. We consider 25,000 different initial conditions generated through PhiFlow [Holl et al., 2020]. We use 22,500 for training and the remaining for testing. We randomly sample the smoke sphere radius $r \sim \mathcal{U}\{5, 30\}$ and the buoyancy coefficient $B \sim \mathcal{U}_{[0.1, 0.5]}$. For each initial condition, we run the simulation for a fixed number of steps and only keep the last state. We sub-sample a $32 \times 32$ patch from each state to construct a dataset. We sample the number of context points according to $N_c \sim \mathcal{U}\{10, 250\}$ and set the remaining datapoints as target points. Table 1 compares the average test log-likelihood of the models. As in the 1-D regression experiment, the approximately equivariant versions of each model outperform both the non-equivariant and equivariant versions, demonstrating the effectiveness of our approach in modelling complex symmetry-breaking features. We provide illustrations of the predictive means for each model in Appendix Figure 6.

## 5.3 Environmental Data

As remarked in Section 1, aspects of modelling climate systems adhere to symmetries such as equivariance. However, there are also unknown local effects which may be revealed by sufficient data. We explore this empirically by considering a real-world dataset derived from ERA5 [Copernicus Climate Change Service, 2020], consisting of surface air temperatures for the years 2018 and 2019. Measurements are collected at a latitudinal and longitudinal resolution of $0.5°$, and temporal resolution of an hour. We also have access to the surface elevation at each coordinate, resulting in a 4-D input ($\mathbf{x}_n \in \mathbb{R}^4$). We consider measurements collected from Europe and from central US.[5]. We train each model on Europe's 2018 data, and test on both Europe's and central US' 2019 data. Because the CNN-based architectures have insufficient support for 4-D convolutions, we first consider a 2-D experiment in which the inputs consist of longitudes and latitudes, followed by a 4-D experiment consisting of all four inputs upon which the transformer-based architectures are evaluated.

**2-D Spatial Regression.** We sample datasets spanning $16°$ across each axis. Each dataset consists of a maximum of $N = 1024$ datapoints, from which the number of context points are sampled according to $N_c \sim \mathcal{U}\{\lfloor N/100 \rfloor, \lfloor N/3 \rfloor\}$. The remaining datapoints are set as target points. Table 2 presents the average test log-likelihood on the two regions for each model. We observe that approximately equivariant models outperform their equivariant counterparts when tested on the same geographical region as the training data. As the central US data falls outside the geographical region of the training data, we zero-out the fixed inputs, so that the predictions for the approximately equivariant models depend solely on their equivariant component. Surprisingly, they also outperform their equivariant counterparts. This suggests that incorporating approximate equivariance acts to regularise the equivariant component of the model, improving generalisation in finite-data settings such as this. In Figure 2, we compare the predictions of the PT-TNP and EquivCNP-based models for a single test dataset with the ground-truth data and the equivariant predictions made by the same models with the fixed inputs zeroed out. The predictive means of both models are almost indistinguishable from ground-truth. We can gain valuable insight into the effectiveness of our approach by comparing a plot of the difference between the approximately equivariant and the equivariant predictions (Figures 2f and 2i) to that of the elevation map for this region (Figure 2c). As elevation is not provided as an input, yet is crucial in predicting surface air temperature, the approximately equivariant models can infer the effect of local topographical features on air temperature through their non-equivariant component. As with the 1-D GP experiment, we analyse the degree of approximate equivariance and the performance with an increasing number of fixed inputs in Appendix E.3, drawing similar conclusions.

**4-D Spatio-Temporal Regression.** In this experiment, we sample datasets across 4 days with measurements every day and spanning $8°$ across each axis. Each dataset consists of a maximum of $N = 1024$ datapoints, from which the number of context points are sampled according to $N_c \sim \mathcal{U}\{\lfloor N/100 \rfloor, \lfloor N/3 \rfloor\}$. The remaining datapoints are set as target points. We provide results in Table 2. Similar to the 2-D experiment, we observe that the approximately equivariant PT-TNP outperforms both the equivariant PT-TNP and non-equivariant PT-TNP on both testing regions.

---

[5]A longitude / latitude range of $[35°, 60°]$ / $[10°, 45°]$ and $[-120°, -80°]$ / $[30°, 50°]$, respectively.

**Table 2:** Average test log-likelihoods (↑) for the 2-D and 4-D environmental regression experiment. Results are grouped together by model class. Best in-class result is bolded.

| Model | 2-D Regression | | 4-D Regression | |
|---|---|---|---|---|
| | Europe (↑) | US (↑) | Europe (↑) | US (↑) |
| PT-TNP | $1.14 \pm 0.01$ | $< -10^6$ | $0.94 \pm 0.01$ | $< -10^{26}$ |
| PT-TNP ($T$) | $1.06 \pm 0.01$ | $\mathbf{0.55 \pm 0.01}$ | $1.14 \pm 0.01$ | $0.73 \pm 0.01$ |
| PT-TNP ($\widetilde{T}$) | $\mathbf{1.22 \pm 0.01}$ | $\mathbf{0.55 \pm 0.01}$ | $\mathbf{1.21 \pm 0.01}$ | $\mathbf{0.76 \pm 0.01}$ |
| ConvCNP ($T$) | $1.11 \pm 0.01$ | $0.12 \pm 0.02$ | - | - |
| ConvCNP ($\widetilde{T}$) | $\mathbf{1.18 \pm 0.01}$ | $0.15 \pm 0.02$ | - | - |
| RelaxedConvCNP ($\widetilde{T}$) | $\mathbf{1.20 \pm 0.01}$ | $\mathbf{0.22 \pm 0.02}$ | - | - |
| EquivCNP ($E$) | $1.27 \pm 0.01$ | $0.64 \pm 0.02$ | - | - |
| EquivCNP ($\widetilde{E}$) | $\mathbf{1.36 \pm 0.01}$ | $\mathbf{0.69 \pm 0.01}$ | - | - |

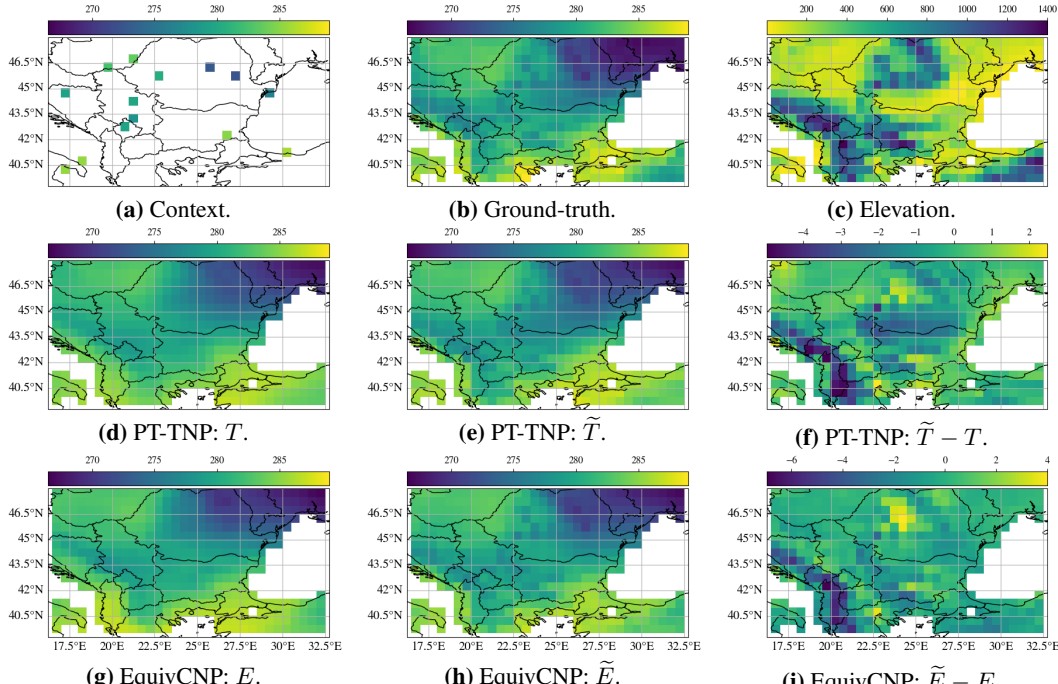

(a) Context.

(b) Ground-truth.

(c) Elevation.

(d) PT-TNP: $T$.

(e) PT-TNP: $\widetilde{T}$.

(f) PT-TNP: $\widetilde{T} - T$.

(g) EquivCNP: $E$.

(h) EquivCNP: $\widetilde{E}$.

(i) EquivCNP: $\widetilde{E} - E$.

**Figure 2:** A comparison between the predictive distributions of the equivariant (left column) and approximately equivariant (middle column) components of the PT-TNP ($\widetilde{T}$) and EquivCNP ($\widetilde{E}$) models on a single (cropped) test dataset from the 2-D environmental data experiment.

## 6 Conclusion

The contributions of this paper are two-fold. First, we develop novel theoretical results that provide insights into the general construction of approximately equivariant operators which is agnostic to the choice of symmetry group and choice of model architecture. Second, we use these insights to construct approximately equivariant NPs, demonstrating their improved performance relative to non-equivariant and strictly equivariant counterparts on a number of synthetic and real-world regression problems. We consider this work to be an important step towards understanding and developing approximately equivariant models. However, more must be done to rigorously quantify and control the degree to which these models depart from strict equivariance. Further, we only considered simple approaches to incorporating approximate equivariance into equivariant architectures, and provided empirical results for relatively small-scale experiments. We look forward to addressing each of these limitations in future work.

## Acknowledgements

CD is supported by the Cambridge Trust Scholarship. AW acknowledges support from a Turing AI fellowship under grant EP/V025279/1 and the Leverhulme Trust via CFI. RET is supported by The Alan Turing Institute, Google, Amazon, ARM, Improbable, EPSRC grant EP/T005386/1, and the EPSRC Probabilistic AI Hub (ProbAI, EP/Y028783/1).

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

# A Proof of Theorem 3

**Proposition 3.** *Let $T \in B$. Then $T$ is compact if and only if $\|T - P_n T P_n\| \to 0$.*

*Proof.* Assume that $\|T - P_n T P_n\| \to 0$. Then $T$ is the limit of a sequence of finite-rank operators, so $T$ is compact (Proposition 2).

On the other hand, assume that $T$ is compact. Then $\|T - P_n T\| \to 0$ (Proposition 2). Use the triangle inequality to bound

$$\|T - P_n T P_n\| = \|T - P_n T + P_n T - P_n T P_n\| \leq \|T - P_n T\| + \|P_n\| \|T - T P_n\|. \quad (9)$$

We already know that $\|T - P_n T\| \to 0$, and it is true that $\|P_n\| = 1$. Finally, since $T$ is compact, the adjoint $T^*$ is also compact. Hence, again by Proposition 2,

$$\|T^* - P_n T^*\| = \|(T - T P_n)^*\| = \|T - T P_n\| \to 0, \quad (10)$$

which proves that $\|T - P_n T P_n\| \to 0$. $\qquad \square$

**Theorem 3** (Approximation of non-equivariant operators.)**.** *Let $T \colon \mathbb{H} \to \mathbb{H}$ be a continuous, possibly nonlinear operator. Assume that $\|T - P_n T P_n\| \to 0$, and that $T$ is $(c, \alpha)$-Hölder for $c, \alpha > 0$, in the following sense:*

$$\|T(u) - T(v)\| \leq c \|u - v\|^{\alpha} \quad \text{for all } u, v \in \mathbb{H}. \quad (5)$$

*Moreover, assume that the orthonormal basis $(e_i)_{i \geq 1}$ is chosen such that, for every $n \in \mathbb{N}$ and $g \in G$, $\operatorname{span}\{e_1, \ldots, e_n\} = \operatorname{span}\{g e_1, \ldots, g e_n\}$, meaning that subspaces spanned by finitely many basis elements are invariant under the group action $(*)$. Let $M > 0$. Then there exists a sequence $(k_n)_{n \geq 1} \subseteq \mathbb{N}$, a sequence of continuous nonlinear operators $E_n \colon \mathbb{H}^{1+k_n} \to \mathbb{H}$, $n \geq 1$, and a sequence of functions $(t_n)_{n \geq 1} \subseteq \mathbb{H}$ such that every $E_n$ is $G$-equivariant,*

$$E_n(g f_1, \ldots, g f_{1+k_n}) = g E_n(f_1, \ldots, f_{1+k_n}) \quad \text{for all } g \in G \text{ and } f_1, \ldots, f_{1+k_n} \in \mathbb{H}, \quad (6)$$

*and*

$$\sup_{u \in \mathbb{H} : \|u\| \leq M} \|T(u) - E_n(u, t_1, \ldots, t_{k_n})\| \to 0. \quad (7)$$

*If assumption $(*)$ does not hold, then the conclusion holds with $E_n(\,\cdot\,, t_1, \ldots, t_{k_n})$ replaced by $E_n(P_n \,\cdot\,, t_1, \ldots, t_{k_n})$. We provide a proof in Appendix A.*

*Proof.* Let $\varepsilon > 0$. Choose $n \in \mathbb{N}$ such that $\|T - P_n T P_n\| < \frac{1}{2}\varepsilon$, and choose $h > 0$ such that $c h^{\alpha} = \frac{1}{2}\varepsilon$. Set $h_n = h/\sqrt{2n}$. Consider the following collection of vectors:

$$A = \{j h_n e_1 : j = 0, \pm 1, \ldots, \pm \lceil M/h_n \rceil\} \times \cdots \times \{j h_n e_n : j = 0, \pm 1, \ldots, \pm \lceil M/h_n \rceil\}. \quad (11)$$

By construction of $A$, for every $u \in \mathbb{H}$ such that $\|u\| \leq M$, there exists an $a \in A$ such that $\|P_n u - a\|^2 \leq n h_n^2 = \frac{1}{2} h^2 < h^2$.

Let $\tilde{k} \colon \mathbb{R} \to [0, \infty)$ be a continuous function with support equal to $(-h, h)$. Set $k \colon \mathbb{H} \times \mathbb{H} \to [0, \infty)$, $k(u, v) = \tilde{k}(\|u - v\|)$. Consider the following map:

$$E \colon \mathbb{H} \times \mathbb{H}^{2|A|} \to \mathbb{H}, \quad E(u, a_1, t_1, \ldots, a_{|A|}, t_{|A|}) = \frac{\sum_{i=1}^{|A|} k(u, a_i) t_i}{\sum_{i=1}^{|A|} k(u, a_i)} \quad (12)$$

where $0/0$ is defined as $0$. This map $E$ is continuous: the numerator is a continuous $\mathbb{H}$-valued function, and the denominator is continuous $\mathbb{R}$-valued function which is non-zero wherever the numerator is non-zero. Moreover, by $G$-invariance of $\mathbb{H}$, $k$ is $G$-invariant, so $E$ is $G$-equivariant:

$$E(gu, g a_1, g t_1, \ldots) = \frac{\sum_{i=1}^{|A|} k(gu, g a_i) g t_i}{\sum_{i=1}^{|A|} k(gu, g a_i)} = g \frac{\sum_{i=1}^{|A|} k(u, a_i) t_i}{\sum_{i=1}^{|A|} k(u, a_i)} = g E(u, a_1, t_1, \ldots). \quad (13)$$

Now set $t_i = (P_n T P_n)(a_i)$. Consider some $u \in \mathbb{H}$ such that $\|u\| \leq M$. By construction of $A$, $\sum_{i=1}^{|A|} k(P_n u, a_i) > 0$. Therefore, by $(c, \alpha)$-Hölder continuity of $T$,

$$\|(P_n T P_n)(u) - E(P_n u, a_1, t_1, \dots)\|$$

$$= \left\| \frac{\sum_{i=1}^{|A|} k(P_n u, a_i)(P_n T P_n)(u)}{\sum_{i=1}^{|A|} k(P_n u, a_i)} - \frac{\sum_{i=1}^{|A|} k(P_n u, a_i)(P_n T P_n)(a_i)}{\sum_{i=1}^{|A|} k(P_n u, a_i)} \right\| \tag{14}$$

$$= \left\| \frac{\sum_{i=1}^{|A|} k(P_n u, a_i)((P_n T P_n)(u) - (P_n T P_n)(a_i))}{\sum_{i=1}^{|A|} k(P_n u, a_i)} \right\| \tag{15}$$

$$\overset{\text{(i)}}{\leq} \frac{\sum_{i=1}^{|A|} k(P_n u, a_i)\|(P_n T P_n)(u) - (P_n T P_n)(a_i)\|}{\sum_{i=1}^{|A|} k(P_n u, a_i)} \tag{16}$$

$$\overset{\text{(ii)}}{\leq} \frac{\sum_{i=1}^{|A|} k(P_n u, a_i) c \|P_n u - a_i\|^\alpha}{\sum_{i=1}^{|A|} k(P_n u, a_i)} \overset{\text{(iii)}}{\leq} c h^\alpha = \tfrac{1}{2}\varepsilon \tag{17}$$

where (i) uses the triangle inequality; (ii) $(P_n T P_n)(u) = (P_n T P_n)(P_n u)$ and $(c, \alpha)$-Hölder continuity of $P_n T P_n$; and (iii) that $\|P_n u - a_i\| \geq h$ implies that $k(P_n u, a_i) = 0$.

Therefore, for all $u \in \mathbb{H}$ such that $\|u\| \leq M$,

$$\|T(u) - E(P_n u, \dots)\| \leq \|T(u) - (P_n T P_n)(u)\| + \|(P_n T P_n)(u) - E(P_n u, \dots)\| \leq \varepsilon. \tag{18}$$

Finally, if condition $(*)$ holds, then $P_n$ is $G$-equivariant, which we show now. Let $u \in \mathbb{H}$ and consider $g \in G$. Since $\text{span}\{e_1, \dots, e_n\} = \text{span}\{ge_1, \dots, ge_n\}$ and $ge_1, \dots, ge_n$ forms an orthonormal basis for span, $\{e_1, \dots, e_n\}$, we can also write $P_n = \sum_{i=1} ge_i \langle ge_i, \cdot \rangle$, so

$$P_n(gu) = \sum_{i=1}^{n} ge_i \langle ge_i, gu \rangle = g \sum_{i=1}^{n} e_i \langle e_i, u \rangle = g(P_n u). \tag{19}$$

Therefore, in the case that condition $(*)$ holds, $P_n$ can be absorbed into the definition of $E_n$. $\qquad\square$

From the proof, we find that $k_n = 2|A| = 2(1 + 2\lceil \sqrt{2n}M/h \rceil)^n \leq 2(2 + 2\sqrt{2n}M/h)^n$ and $c h^\alpha = \tfrac{1}{2}\varepsilon$, which implies that $h = (\varepsilon/2c)^{1/\alpha}$, so $k_n \leq 2(2 + 2\sqrt{2n}M(\varepsilon/2c)^{-1/\alpha})^n$. This growth estimate is faster than exponential. Like the linear case shows (Theorem 2), better estimates may be obtained with constructions of $E$ that better exploit structure of $T$.

## B Equivariant Neural Processes

In this section, we outline the equivariant NP models we use throughout our experiments, and how their approximately equivariant counterparts can easily be constructed using the results from Section 3. In all cases, the asymptotic space and time complexity remains unchanged.

### B.1 Convolutional Conditional Neural Process

The ConvCNP [Gordon et al., 2019] is a translation equivariant NP and is constructed in exactly the form shown in Equation 2, with $\psi : \mathcal{X}^2 \to \mathbb{R}$ an RBF kernel with learnable lengthscale and $\phi(\mathbf{y}_i) = [\mathbf{y}_i^T; \mathbf{1}]^T$. The functional embedding is discretised and passed pointwise through an MLP to some final functional representation $\tilde{e}(\mathcal{D}) : \tilde{\mathcal{X}} \to \mathbb{R}^{D_z}$, where $\tilde{\mathcal{X}}$ denotes the discretised input domain. $\rho$ implemented as a CNN through which $\tilde{e}(\mathcal{D})$ is passed through together with another RBF kernel $\psi_p$ which maps back to a continuous function space. We provide pseudo-code for a forward pass through the ConvCNP in Algorithm 1.

**Algorithm 1:** Forward pass through the Con-vCNP ($T$) for off-the-grid data.

---

**Input:** $\rho = (\text{CNN}, \psi_p)$, $\psi$, and density $\gamma$.
Context
$\mathcal{D}_c = (\mathbf{X}_c, \mathbf{Y}_c) = \{(\mathbf{x}_{c,n}, \mathbf{y}_{c,n})\}_{n=1}^{N_c}$,
and target $\mathbf{X}_t = \{\mathbf{x}_{t,m}\}_{m=1}^{N_t}$

**begin**
    lower, upper $\leftarrow$ range $(\mathbf{X}_t \cup \mathbf{X}_c)$;
    $\{\tilde{\mathbf{x}}_i\}_{i=1}^T \leftarrow$ grid(lower, upper, $\gamma$);
    $\mathbf{h}_i \leftarrow \sum_{n=1}^{N_c} [1, \mathbf{y}_{c,n}^T]^T \psi(\tilde{\mathbf{x}}_i - \mathbf{x}_{c,n})$;
    $\mathbf{h}_i^{(1)} \leftarrow \mathbf{h}_i^{(1)}/\mathbf{h}_i^{(0)}$;
    $\mathbf{h}_i \leftarrow \text{MLP}(\mathbf{h}_i)$;
    $\{f(\tilde{\mathbf{x}}_i)\}_{i=1}^T \leftarrow \text{CNN}\left(\{\tilde{\mathbf{x}}_i, \mathbf{h}_i\}_{i=1}^T\right)$;
    $\theta_m \leftarrow \sum_{i=1}^T f(\tilde{\mathbf{x}}_i)\psi_p(\mathbf{x}_{t,m} - \tilde{\mathbf{x}}_i)$;
    **return** $p(\cdot|\theta_m)$;
**end**

**Algorithm 2:** Forward pass through the Con-vCNP ($\widetilde{T}$) for off-the-grid data.

---

**Input:** $\rho = (\text{CNN}, \psi_p)$, $\psi$, density $\gamma$, and fixed
input $t$. Context
$\mathcal{D}_c = (\mathbf{X}_c, \mathbf{Y}_c) = \{(\mathbf{x}_{c,n}, \mathbf{y}_{c,n})\}_{n=1}^{N_c}$,
and target $\mathbf{X}_t = \{\mathbf{x}_{t,m}\}_{m=1}^{N_t}$

**begin**
    lower, upper $\leftarrow$ range $(\mathbf{X}_t \cup \mathbf{X}_c)$;
    $\{\tilde{\mathbf{x}}_i\}_{i=1}^T \leftarrow$ grid(lower, upper, $\gamma$);
    $\mathbf{h}_i \leftarrow \sum_{n=1}^{N_c} [1, \mathbf{y}_{c,n}^T]^T \psi(\tilde{\mathbf{x}}_i - \mathbf{x}_{c,n})$;
    $\mathbf{h}_i^{(1)} \leftarrow \mathbf{h}_i^{(1)}/\mathbf{h}_i^{(0)}$;
    $\mathbf{h}_i \leftarrow \text{MLP}(\mathbf{h}_i) + t(\tilde{\mathbf{x}}_i)$;
    $\{f(\tilde{\mathbf{x}}_i)\}_{i=1}^T \leftarrow \text{CNN}\left(\{\tilde{\mathbf{x}}_i, \mathbf{h}_i\}_{i=1}^T\right)$;
    $\theta_m \leftarrow \sum_{i=1}^T f(\tilde{\mathbf{x}}_i)\psi_p(\mathbf{x}_{t,m} - \tilde{\mathbf{x}}_i)$;
    **return** $p(\cdot|\theta_m)$;
**end**

To achieve approximate translation equivariance, we construct a learnable function $t : \tilde{\mathcal{X}} \to \mathbb{R}^{D_z}$ using an MLP which represents the fixed inputs. We sum these together with the resized functional embedding, resulting in the overall implementation $\rho(e(\mathcal{D}), t_1, \ldots, t_B) = \text{CNN}(\tilde{e}(\mathcal{D}) + t)$. Note that the number of fixed inputs is $B = D_z$. We provide pseudo-code for a forward pass through the approximately translation equivariant ConvCNP in Algorithm 2.

## B.2 Equivariant Conditional Neural Process

The EquivCNP [Kawano et al., 2021] is a generalisation of the ConvCNP to more general group equivariances. As with the ConvCNP, it is constructed in the form shown in Equation 2. We achieve $E$-equivariance with $\psi : \mathcal{X}^2 \to \mathbb{R}$ an RBF kernel with learnable lengthscale and $\phi(\mathbf{y}_i) = [\mathbf{y}_i^T; 1]^T$. The functional embedding is discretised and passed pointwise through an MLP to some final functional representation $\tilde{e}(\mathcal{D}) : \tilde{\mathcal{X}} \to \mathbb{R}^{D_z}$, where $\tilde{\mathcal{X}}$ denotes the discretised input domain. $\rho$ is implemented as a group equivariant CNN [Cohen and Welling, 2016a] together with another RBF kernel $\psi_p$ which maps back to a continuous function space. The approximately $E$-equivariant EquivCNP is constructed in an analogous manner to the approximately translation equivariant ConvCNP.

## B.3 Translation Equivariant Transformer Neural Process

The TE-TNP [Ashman et al., 2024] is a translation equivariant NP. However, unlike the ConvCNP, the function embedding representation differs slightly to that in Equation 2, and is given by

$$e(\mathcal{D})(\cdot) : \mathcal{X} \to \mathbb{R}^{D_z} = \text{cat}\left(\{\sum_{i=1}^N \alpha_h(\mathbf{z}_0, \phi(\mathbf{y}_i), \cdot, \mathbf{x}_i)\phi(\mathbf{y}_i)^T \mathbf{W}_{V,h}\}_{h=1}^H\right)\mathbf{W}_0 \quad (20)$$

where

$$\alpha_h(\mathbf{z}_0, \phi(\mathbf{y}_i), \cdot, \mathbf{x}_i) = \frac{e^{\mu(\mathbf{z}_0^T \mathbf{W}_{Q,h}[\mathbf{W}_{K,h}]^T \phi(\mathbf{y}_i), \cdot - \mathbf{x}_i)}}{\sum_{j=1}^N e^{\mu(\mathbf{z}_0^T \mathbf{W}_{Q,h}[\mathbf{W}_{K,h}]^T \phi(\mathbf{y}_j), \cdot - \mathbf{x}_j)}}. \quad (21)$$

Here, $\mu$ is implemented using an MLP. This is a partially evaluated translation equivariant multi-head cross-attention (TE-MHCA) operation, with attention weights computed according to Equation 21. We refer to $\phi(\mathbf{y}_i)$ as the initial token embedding for $\mathbf{y}_i$. $\rho$ is implemented using translation equivariant multi-head self-attention (TE-MHSA) operations which update the token representations. These are combined with TE-MHCA operations which map the token representations to a continuous function space. We provide pseudo-code for a forward pass through the TE-TNP in Algorithm 3.

**Algorithm 3:** Forward pass through the TE-TNP ($T$).

**Input:** $\rho = \{\text{TE-MHSA}^{(\ell)}, \text{TE-MHSA}^{(\ell)}\}_{\ell=1}^{L}$, $\phi$. Context $\mathcal{D}_c = (\mathbf{X}_c, \mathbf{Y}_c) = \{(\mathbf{x}_{c,n}, \mathbf{y}_{c,n})\}_{n=1}^{N_c}$, and target $\mathbf{X}_t = \{\mathbf{x}_{t,m}\}_{m=1}^{N_t}$

**begin**

> $\mathbf{z}_{c,n} \leftarrow \phi(\mathbf{y}_{c,n})$;
> $\mathbf{z}_{t,m} \leftarrow \mathbf{z}_0$;
> **for** $\ell = 1, \ldots, L$ **do**
>> $\mathbf{z}_{t,m} \leftarrow$ TE-MHCA$^{(\ell)}(\mathbf{z}_{t,m}, \mathbf{Z}_c, \mathbf{x}_{t,m}, \mathbf{X}_c)$;
>> $\{\mathbf{z}_{c,n}\}_{n=1}^{N_c} \leftarrow$ TE-MHSA$^{(\ell)}(\mathbf{Z}_c, \mathbf{X}_c)$;
> **end**
> $\theta_m \leftarrow \text{MLP}(\mathbf{z}_{t,m})$;
> **return** $p(\cdot|\theta_m)$;

**end**

**Algorithm 4:** Forward pass through the TE-TNP ($\widetilde{T}$).

**Input:** $\rho = \{\text{TE-MHSA}^{(\ell)}, \text{TE-MHSA}^{(\ell)}\}_{\ell=1}^{L}$, $\phi$, fixed input $t$. Context $\mathcal{D}_c = (\mathbf{X}_c, \mathbf{Y}_c) = \{(\mathbf{x}_{c,n}, \mathbf{y}_{c,n})\}_{n=1}^{N_c}$, and target $\mathbf{X}_t = \{\mathbf{x}_{t,m}\}_{m=1}^{N_t}$

**begin**

> $\mathbf{z}_{c,n} \leftarrow \phi(\mathbf{y}_{c,n}) + t(\mathbf{x}_{c,n})$;
> $\mathbf{z}_{t,m} \leftarrow \mathbf{z}_0$;
> **for** $\ell = 1, \ldots, L$ **do**
>> $\mathbf{z}_{t,m} \leftarrow$ TE-MHCA$^{(\ell)}(\mathbf{z}_{t,m}, \mathbf{Z}_c, \mathbf{x}_{t,m}, \mathbf{X}_c)$;
>> $\{\mathbf{z}_{c,n}\}_{n=1}^{N_c} \leftarrow$ TE-MHSA$^{(\ell)}(\mathbf{Z}_c, \mathbf{X}_c)$;
> **end**
> $\theta_m \leftarrow \text{MLP}(\mathbf{z}_{t,m})$;
> **return** $p(\cdot|\theta_m)$;

**end**

Unlike the ConvCNP and EquivCNP, we do not discretise the input domain of the functional representation. It is not clear how one would sum together a fixed input and the functional embedding without requiring infinite summations over the entire input domain. Thus, we take an alternative approach in which the initial token representations for context set are modified by summing together with the fixed input value at the corresponding location. This modification still conforms to the form given in Equation 8, and is simple to implement. We provide pseudo-code for a forward pass through the approximately translation equivariant TE-TNP in Algorithm 4.

### B.4 Pseudo-Token Translation Equivariant Transformer Neural Process

The pseudo-token TE-TNP (PT-TE-TNP) is an alternative to the TE-TNP which avoids the $\mathcal{O}(N^2)$ computational cost of regular transformers through the use of pseudo-tokens [Feng et al., 2022, Ashman et al., 2024, Lee et al., 2019]. The functional embedding representation for the PT-TE-TNP is given by

$$e(\mathcal{D})(\cdot) : \mathcal{X} \rightarrow \mathbb{R}^{D_z} = \text{cat}\left(\{\sum_{m=1}^{M} \alpha_h(\mathbf{z}_0, \mathbf{u}_m, \cdot, \mathbf{x}_i)\mathbf{u}_m^T \mathbf{W}_{V,h}\}_{h=1}^{H}\right)\mathbf{W}_0 \tag{22}$$

where

$$\mathbf{u}_m = \text{cat}\left(\{\sum_{i=1}^{N} \alpha_h(\mathbf{u}_{m,0}, \phi(\mathbf{y}_i), \mathbf{v}_m, \mathbf{x}_i)\phi(\mathbf{y}_i)^T \mathbf{W}_{V,h}\}_{h=1}^{H}\right)\mathbf{W}_0 \tag{23}$$

and

$$\mathbf{v}_m = \mathbf{v}_{m,0} + \frac{1}{N}\sum_{i=1}^{N} \mathbf{x}_i. \tag{24}$$

The attention mechanism used in the PT-TE-TNP is the same as in the TE-TNP, and is given in Equation 21. Intuitively, the initial token embeddings for the dataset $\mathcal{D}$ are 'summarised' by the pseudo-tokens $\{\mathbf{u}_m\}_{m=1}^{M}$ and pseudo-token input locations $\{\mathbf{v}_m\}_{m=1}^{M}$. $\rho$ is implemented using a series of TE-MHSA and TE-MHCA operations which update the pseudo-tokens and form a mapping from pseudo-tokens to a continuous function space. We provide pseudo-code for a forward pass through the PT-TE-TNP in Algorithm 5. Note that this assumes that the perceiver-style approach is taken [Jaegle et al., 2021], as in the latent bottlenecked attentive NP of Feng et al. [2022]. The induced set transformer approach of Lee et al. [2019] can also be used.

**Algorithm 5:** Forward pass through the PT-TE-TNP ($T$).

**Input:** $\rho =$
$\{\text{TE-MHSA}^{(\ell)}, \text{TE-MHCA}_1^{(\ell)}, \text{TE-MHCA}_2^{(\ell)}\}_{\ell=1}^L$,
$\phi$, $\mathbf{V} = \{\mathbf{v}_{m,0}\}_{j=1}^M$, $\mathbf{U} = \{\mathbf{u}_{j,0}\}_{m=1}^M$, $\mathbf{z}_0$.
Context
$\mathcal{D}_c = (\mathbf{X}_c, \mathbf{Y}_c) = \{(\mathbf{x}_{c,n}, \mathbf{y}_{c,n})\}_{n=1}^{N_c}$,
and target $\mathbf{X}_t = \{\mathbf{x}_{t,m}\}_{m=1}^{N_t}$

**begin**
> $\mathbf{z}_{c,n} \leftarrow \phi(\mathbf{y}_{c,n})$;
> $\mathbf{v}_m \leftarrow \mathbf{v}_{m,0} + 1/N \sum_{i=1}^{N_c} \mathbf{x}_{c,i}$;
> $\mathbf{u}_j \leftarrow \mathbf{u}_{j,0}$;
> $\mathbf{z}_{t,m} \leftarrow \mathbf{z}_0$;
> **for** $\ell = 1, \ldots, L$ **do**
> > $\mathbf{u}_j \leftarrow \text{TE-MHCA}_1^{(\ell)}(\mathbf{u}_j, \mathbf{Z}_c, \mathbf{v}_j, \mathbf{X}_c)$;
> > $\{\mathbf{u}_j\}_{j=1}^M \leftarrow \text{TE-MHSA}^{(\ell)}(\mathbf{U}, \mathbf{V})$;
> > $\mathbf{z}_{t,m} \leftarrow$
> > $\quad \text{TE-MHCA}_2^{(\ell)}(\mathbf{z}_{t,m}, \mathbf{U}, \mathbf{z}_{t,m}, \mathbf{V})$;
> **end**
> $\theta_m \leftarrow \text{MLP}(\mathbf{z}_{t,m})$;
> **return** $p(\cdot|\theta_m)$;

**end**

---

**Algorithm 6:** Forward pass through the PT-TE-TNP ($\widetilde{T}$).

**Input:** $\rho =$
$\{\text{TE-MHSA}^{(\ell)}, \text{TE-MHCA}_1^{(\ell)}, \text{TE-MHCA}_2^{(\ell)}\}_{\ell=1}^L$,
$\phi$, $\mathbf{V} = \{\mathbf{v}_{m,0}\}_{j=1}^M$, $\mathbf{U} = \{\mathbf{u}_{j,0}\}_{m=1}^M$, $\mathbf{z}_0$.
Context
$\mathcal{D}_c = (\mathbf{X}_c, \mathbf{Y}_c) = \{(\mathbf{x}_{c,n}, \mathbf{y}_{c,n})\}_{n=1}^{N_c}$,
and target $\mathbf{X}_t = \{\mathbf{x}_{t,m}\}_{m=1}^{N_t}$

**begin**
> $\mathbf{z}_{c,n} \leftarrow \phi(\mathbf{y}_{c,n})$;
> $\mathbf{v}_m \leftarrow \mathbf{v}_{m,0} + 1/N \sum_{i=1}^{N_c} \mathbf{x}_{c,i}$;
> $\mathbf{u}_j \leftarrow \mathbf{u}_{j,0} + t(\mathbf{v}_m)$;
> $\mathbf{z}_{t,m} \leftarrow \mathbf{z}_0$;
> **for** $\ell = 1, \ldots, L$ **do**
> > $\mathbf{u}_j \leftarrow \text{TE-MHCA}_1^{(\ell)}(\mathbf{u}_j, \mathbf{Z}_c, \mathbf{v}_j, \mathbf{X}_c)$;
> > $\{\mathbf{u}_j\}_{j=1}^M \leftarrow \text{TE-MHSA}^{(\ell)}(\mathbf{U}, \mathbf{V})$;
> > $\mathbf{z}_{t,m} \leftarrow$
> > $\quad \text{TE-MHCA}_2^{(\ell)}(\mathbf{z}_{t,m}, \mathbf{U}, \mathbf{z}_{t,m}, \mathbf{V})$;
> **end**
> $\theta_m \leftarrow \text{MLP}(\mathbf{z}_{t,m})$;
> **return** $p(\cdot|\theta_m)$;

**end**

---

We incorporate the fixed inputs by modifying summing together the initial pseudo-token values with the fixed inputs evaluated at the corresponding pseudo-token input location. We provide pseudo-code for a forward pass through the approximately translation equivariant PT-TE-TNP in Algorithm 6.

### B.5 Relaxed Convolutional Conditional Neural Process

The RelaxedConvCNP is equivalent to the ConvCNP with the RelaxedCNN of Wang et al. [2022a] used in place of a standard CNN. The RelaxedCNN replaces standard convolutions with relaxed convolutions, which we discuss in more detail in Appendix C. In short, the relaxed convolution is defined as

$$(k \stackrel{*}{*} f)(u) = \int_G \sum_{l=1}^L f(v) w_l(v) k_l(u^{-1} v) d\mu(v). \tag{25}$$

This modifies the kernel weights $k_l(u^{-1}v)$ with the input-dependent function $w_l(v)$, which is our fixed input. To enable the fixed inputs to be zeroed out, we modify this slightly as

$$(k \stackrel{\sim}{*} f)(u) = \int_G \sum_{l=1}^L f(v)(1 + t_l(v)) k_l(u^{-1} v) d\mu(v) \tag{26}$$

so that when $t_l(v) = 0$ we recover the standard $G$-equivariant convolution. We make $t_l(v)$ a learnable function parameterised by an MLP, as with the other approaches.

## C  Unification of Existing Approximately Equivariant Architectures

In this section, we demonstrate that the input-dependent kernel approaches of Wang et al. [2022a] and van der Ouderaa et al. [2022] are special cases of our approach. Throughout this section, we consider scalar-valued functions. We define the action of $g \in G$ on $f : G \to \mathbb{R}$ as

$$(g \cdot f)(u) = f(g^{-1} u). \tag{27}$$

We begin with the approach of Wang et al. [2022a], which defines the relaxed group convolution as

$$(k \stackrel{\sim}{*} f)(u) = \int_G \sum_{l=1}^L f(v) w_l(v) k_l(u^{-1} v) d\mu(v) \tag{28}$$

where $\mu(v)$ is the left Haar measure on $G$. Observe that this replaces a single kernel with a set of kernels $\{k_l\}_{l=1}^L$, whose contributions are linearly combined with coefficients that vary with $v \in G$. This can be expressed as

$$(k \; \tilde{\ast} \; f)(u) = \sum_{l=1}^{L} \underbrace{\int_G f(v)w_l(v)k_l(u^{-1}v)d\mu(v)}_{E_l(f,w_l)(u)} = \sum_{l=1}^{L} E_l(f, w_l) = E(f, w_1, \ldots, w_L)(u) \quad (29)$$

where $w_l$ correspond to the fixed inputs and $E$ is used to denote equivariant operators w.r.t. the group $G$. Clearly each $E_l$ is $G$-equivariant with respect to its inputs, as applying a transformation $g \in G$ to the inputs gives

$$
\begin{aligned}
E_l(g \cdot f, g \cdot w_l)(u) &= \int_G f(g^{-1}v)w_l(g^{-1}v)k_l(u^{-1}v)d\mu(v) \\
&= \int_G f(v)w_l(v)k_l(u^{-1}gv)d\mu(v) \\
&= \int_G f(v)w_l(v)k_l((g^{-1}u)^{-1}v)d\mu(v) \\
&= g \cdot E_l(f, w_l)(u).
\end{aligned}
\quad (30)
$$

Since the sum of $G$-equivariant functions is itself $G$-equivariant, $E(f, w_1, \ldots, w_L)(u)$ is $G$-equivariant with respect to its inputs.

The approach of van der Ouderaa et al. [2022] is more general than that of Wang et al. [2022a], and relax strict equivariance through convolutions with input-dependent kernels:

$$(k \; \tilde{\ast} \; f)(u) = \int_G k(v^{-1}u, v)f(v)d\mu(v). \quad (31)$$

Define a fixed input $t(u) = u$. We can express the above convolution as a $G$-equivariant operation on $t$ and $f$:

$$E(f, t)(u) = \int_G k(v^{-1}u, t(v))f(v)d\mu(v). \quad (32)$$

To demonstrate $G$-equivariance, consider applying a transformation $g \in G$ to the inputs:

$$
\begin{aligned}
E(g \cdot f, g \cdot t)(u) &= \int_G k(v^{-1}u, t(g^{-1}v))f(g^{-1}v)d\mu(v) \\
&= \int_G k(v^{-1}g^{-1}u, t(v))f(v)d\mu(v) \\
&= \int_G k(v^{-1}(g^{-1}u), t(v))f(v)d\mu(v) \\
&= g \cdot E(f, t)(u).
\end{aligned}
\quad (33)
$$

Thus, this method is also a special case of ours.

## D  Achieving $\epsilon$-Approximate $G$-Equivariance

Wang et al. [2022b] introduce a useful metric for assessing the degree to which mappings are approximately equivariant. Specifically, let $\pi \colon X \to Y$ denote some mapping between $G$-spaces $X$ and $Y$. Let $d \colon Y \times Y \to \mathbb{R}$ be a metric on $Y$. We say $\pi$ is $\epsilon$-approximately $G$-equivariant if for all $g \in G$ and $x \in X$

$$d\big(\pi(gx), \; g\pi(x)\big) \leq \epsilon. \quad (34)$$

Let $\pi_{\text{AE-NP}} \colon \mathcal{S} \to \mathcal{C}(\mathcal{X}, \Theta)$ be a learned neural process that breaks equivariance with fixed additional inputs $(t_b)_{b=1}^B$, and let $\pi_{\text{E-NP}} \colon \mathcal{S} \to \mathcal{C}(\mathcal{X}, \Theta)$ be the strictly equivariant neural process obtained by setting $t_b = 0$ in $\pi_{\text{AE-NP}}$. If the predictions of $\pi_{\text{AE-NP}}$ are not too different from those of $\pi_{\text{E-NP}}$, because the fixed additional inputs only affect the predictions in a limited way, then in this appendix we argue that $\pi_{\text{AE-NP}}$ is $\epsilon$-approximately equivariant.

To assume that predictions of $\pi_{\text{AE-NP}}$ are not too different from those of $\pi_{\text{E-NP}}$, assume that there exists some metric $d$ and $\epsilon > 0$ such that, for all $\mathcal{D}$, $d(\pi_{\text{AE-NP}}(\mathcal{D}), \pi_{\text{E-NP}}(\mathcal{D})) \leq \epsilon$. For example, the metric can be the root-mean-squared-error (RMSE) between the mean functions $x \mapsto \mathbb{E}[f(x)]$ of two stochastic processes by integrating over the input domain $\mathcal{X}$. In practice, we will find that $d(\pi_{\text{AE-NP}}(\mathcal{D}), \pi_{\text{E-NP}}(\mathcal{D})) \leq \epsilon$ holds for $\mathcal{D}$ over the training domain. If we assume that $\pi_{\text{AE-NP}}$ has a finite receptive field—in the sense that the output stochastic process has only local dependencies—then, provided we zero out the additional fixed inputs as described in Section 3.1, $d(\pi_{\text{AE-NP}}(\mathcal{D}), \pi_{\text{E-NP}}(\mathcal{D})) \leq \epsilon$ will hold over the entire input domain.

Finally, to show that $\pi_{\text{AE-NP}}$ is $\epsilon$-approximately equivariant, we use the triangle inequality:

$$d\big(g\pi_{\text{AE-NP}}(\mathcal{D}), \pi_{\text{AE-NP}}(g\mathcal{D})\big) \leq \underbrace{d\big(g\pi_{\text{AE-NP}}(\mathcal{D}), g\pi_{\text{E-NP}}(\mathcal{D})\big)}_{\leq \epsilon} + \underbrace{d\big(g\pi_{\text{E-NP}}(\mathcal{D}), \pi_{\text{E-NP}}(g\mathcal{D})\big)}_{= 0}$$
$$+ \underbrace{d\big(\pi_{\text{E-NP}}(g\mathcal{D}), \pi_{\text{AE-NP}}(g\mathcal{D})\big)}_{\leq \epsilon} \tag{35}$$
$$\leq 2\epsilon, \tag{36}$$

where $d\big(g\pi_{\text{E-NP}}(\mathcal{D}), \pi_{\text{E-NP}}(g\mathcal{D})\big) = 0$ because $\pi_{\text{E-NP}}$ is strictly equivariant.

# E   Experiment Details and Additional Results

## E.1   Synthetic 1-D Regression

We consider a synthetic 1-D regression task using samples drawn from Gaussian processes (GPs) with the Gibbs kernel with an observation noise of 0.2. This kernel is a non-stationary generalisation of the squared exponential kernel, where the lengthscale parameter becomes a function of position $l(x)$:

$$k(x, x'; l) = \sqrt{\left(\frac{2l(x)l(x')}{l(x)^2 + l(x')^2}\right)} \exp\left(-\frac{(x - x')^2}{l(x)^2 + l(x')^2}\right)$$

We consider a 1-D space with two regions with constant, but different lengthscale - one with $l = 0.1$, and one with $l = 4.0$. The lengthscale changepoint is situated at $x = 0$. We randomly sample with a 0.5 probability the orientation (left/right) of the low/high lengthscale region. Formally, $l(x) = (0.1\beta + 4(1 - \beta))\delta[x < 0] + (0.1(1 - \beta) + 4\beta)\delta[x \geq 0]$, where $\beta \sim \text{Bern}(0.5)$. For each task, we sample the number of context points $N_c \sim \mathcal{U}\{1, 64\}$ and set the number of target points to $N_t = 128$. The context range $[\mathbf{x}_{c,\text{min}}, \mathbf{x}_{c,\text{max}}]$ (from which the context points are uniformly sampled) is an interval of length 4, with its centre randomly sampled according to $\mathcal{U}_{[-7,7]}$ for the ID task, and according to $\mathcal{U}_{[13,27]}$ for the OOD task. The target range is $[\mathbf{x}_{t,\text{min}}, \mathbf{x}_{t,\text{max}}] = [\mathbf{x}_{c,\text{min}} - 1, \mathbf{x}_{c,\text{max}} + 1]$. This is also applicable during testing, with the test dataset consisting of 80,000 datasets.

We use an embedding / token size of $D_z = 128$ for the TNP-based models and $D_z = 64$ for the ConvCNP-based ones, and a decoder consisting of an MLP with two hidden layers of dimension $D_z$. The decoder parameterises the mean and pre-softplus variance of a Gaussian likelihood with heterogeneous noise. Model specific architectures are as follows:

**TNP** The initial context tokens are obtained by passing the concatenation $[x, y, 1]$ through an MLP with two hidden layers of dimension $D_z$. The initial target tokens are obtained by passing the concatenation $[x, 0, 0]$ through the same MLP. The final dimension of the input acts as a 'density' channel to indicate whether or not an observation is present. The TNP encoder consists of nine layers of self-attention and cross-attention blocks, each with $H = 8$ attention heads with dimensions $D_V = D_{QK} = 16$. In each of the attention blocks, we apply a residual connection consisting of layer-normalisation to the input tokens followed by the attention mechanism. Following this, there is another residual connection consisting of a layer-normalisation followed by a pointwise MLP with two hidden layers of dimension $D_z$.

**TNP ($T$)** For the TNP ($T$) model we follow Ashman et al. [2024]. The architecture is similar to the TNP model, with the attention blocks replaced with their translation equivariant counterparts. For the translation equivariant attention mechanisms, we implement $\rho^\ell : \mathbb{R}^H \times \mathbb{R}^{D_x} \to \mathbb{R}^H$ as an MLP with

two hidden layers of dimension $D_z$. The initial context token embeddings are obtained by passing the context observations through an MLP with two hidden layers of dimension $D_z$. The initial target token embeddings are sampled from a standard normal. Pseudo-code for a forward pass through the TNP ($T$) can be found in Appendix B.3.

**TNP ($\widetilde{T}$)** The architecture of the TNP ($\widetilde{T}$) is similar to that of the TNP ($T$), with the exception of the extra fixed inputs that are added to the context token representation. These are obtained by first performing a Fourier expansion to the context token locations, and then passing the result through an MLP with two hidden layers of dimension $D_z$. We use four Fourier coefficients, zero out the fixed inputs outside of [-7, 7], and during training we drop them out with a probability of 0.5. Pseudo-code for a forward pass through the TNP ($\widetilde{T}$) can be found in Appendix B.3.

**ConvCNP ($T$)** For the ConvCNP model, we use a CNN with 9 layers. We use $C = 64$ channels, a kernel size $k = 21$ with a stride of one. The input domain is discretised with 46 points per unit. The decoder uses five different learned lengthscales to map the output of the CNN back to a continuous function. Pseudo-code for a forward pass through the ConvCNP ($T$) can be found in Appendix B.1.

**ConvCNP ($\widetilde{T}$)** The ConvCNP ($\widetilde{T}$) closely follows the ConvCNP ($T$) model, with the main difference being in the input to the model. For the approximately equivariant model, we sum up the output of the resized functional embedding with the representation of the fixed inputs. The latter is obtained by passing the input locations (that lie on the grid) through and MLP with two hidden layers of dimension $C$. We consider $C$ fixed inputs, we zero them out outside of [-7, 7] and during training we drop them out with a probability of 0.1. Pseudo-code for a forward pass through the ConvCNP ($\widetilde{T}$) can be found in Appendix B.1.

**RelaxedConvCNP ($\widetilde{T}$)** We use an identical architecture to the ConvCNP($T$), with regular convolutional operations replaced by relaxed convolutions. We use $L = 1$ kernels such that the total parameter count remains similar (see Appendix B.5)

**EquivCNP ($E$)** We use an identical architecture to the ConvCNP ($T$), with symmetric convolutions in place of regular convolutions. Pseudo-code for a forward pass through the EquivCNP ($E$) can be found in Appendix B.2.

**EquivCNP ($\widetilde{E}$)** We use an identical architecture to the ConvCNP ($\widetilde{T}$), with symmetric convolutions in place of regular convolutions. Pseudo-code for a forward pass through the EquivCNP ($\widetilde{E}$) can be found in Appendix B.2.

**Training Details and Compute** For all models, we optimise the model parameters using AdamW [Loshchilov and Hutter, 2017] with a learning rate of $5 \times 10^{-4}$ and batch size of 16. Gradient value magnitudes are clipped at 0.5. We train for a maximum of 500 epochs, with each epoch consisting of 16,000 datasets (10,000 iterations per epoch). We evaluate the performance of each model on 80,000 test datasets. We train and evaluate all models on a single 11 GB NVIDIA GeForce RTX 2080 Ti GPU.

**Additional Results** In Figure 1 we compared the predictive distributions of the eight considered models for a test dataset where the context range spanned both the low and high-lengthscale regions. In Figure 3 and Figure 4 we provide additional examples where the context range only spans one region (the low-lengthscale one in Figure 3 and high-lengthscale one in Figure 4).

Figure 3 shows that the non-equivariant model (TNP) does not produce well-calibrated uncertainties far away from the context region. The equivariant models underestimate the uncertainty near the change point location $x = 0$, giving rise to overly-confident predictions at the transition between the low and high-lengthscale regions. For the TNP ($T$) the uncertainties remain low throughout the entire high-lengthscale region. In contrast, the approximately equivariant models manage to more accurately capture the uncertainties beyond the transition point, into the high-lengthscale region. This indicates that the approximately equivariant models are better suited to cope with non-stationarities in the data.

When the context region only spans the high-lengthscale region, both the strictly and approximately equivariant models output predictions that closely follow the ground truth. However, the non-equivariant model completely fails to generalise, outputting almost symmetric predictions about the origin ($x = 0$). We hypothesise this is because of its inability to generalise, resulting from the lack of suitable inductive biases.

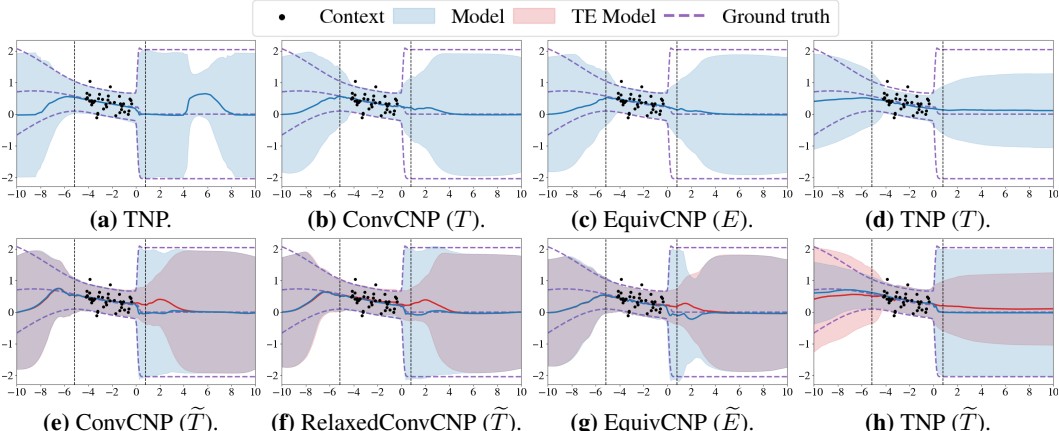

**Figure 3:** A comparison between the predictive distributions on a single synthetic 1D regression dataset of the TNP-, ConvCNP-, and EquivCNP-based models with different inductive biases (non-equivariant, equivariant, or approximately equivariant). Unlike in Figure 1, the context range only spans the low-lengthscale region. For the approximately equivariant models, we plot both the model prediction (blue), as well as the predictions obtained without using the fixed inputs, which results in a strictly equivariant model (red). The approximately equivariant models are the only ones able to correctly capture the uncertainties around the lengthscale change point ($x = 0$).

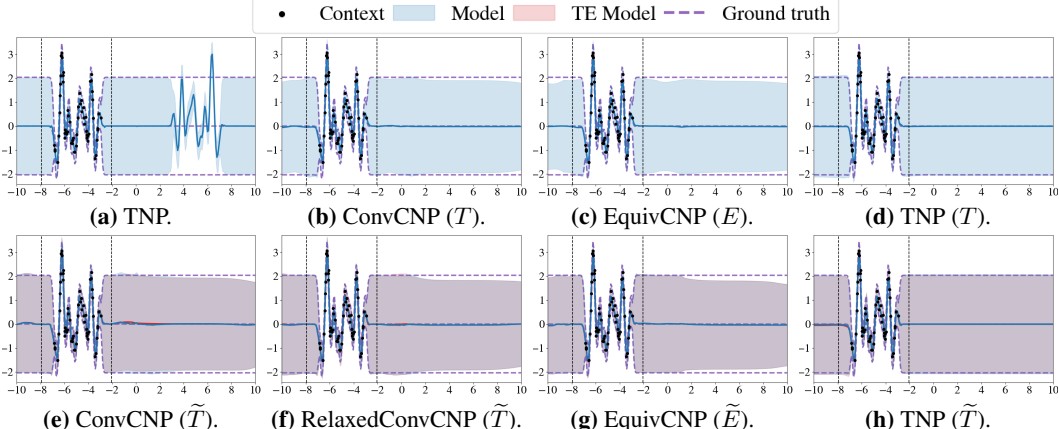

**Figure 4:** A comparison between the predictive distributions on a single synthetic 1D regression dataset of the TNP-, ConvCNP-, and EquivCNP-based models with different inductive biases (non-equivariant, equivariant, or approximately equivariant). The context range only spans the high-lengthscale region. For the approximately equivariant models, we plot both the model prediction (blue), as well as the predictions obtained without using the fixed inputs, which results in a strictly equivariant model (red). Both the strictly and approximately equivariant models output predictions that closely resemble the ground truth, but the non-equivariant TNP model completely fails to generalise.

**Table 3:** Average log-likelihoods (↑) for the synthetic 1-D GP experiment when tested on context sets. Ground truth log-likelihood is $0.2806 \pm 0.0005$.

| Model | Context log-likelihood (↑) |
|---|---|
| TNP | $0.2296 \pm 0.0007$ |
| TNP ($T$) | $0.2396 \pm 0.0013$ |
| TNP ($\widetilde{T}$) | $0.2344 \pm 0.0020$ |
| ConvCNP ($T$) | $0.2362 \pm 0.0007$ |
| ConvCNP ($\widetilde{T}$) | $0.2218 \pm 0.0009$ |
| RelaxedConvCNP ($\widetilde{T}$) | $0.2381 \pm 0.0008$ |
| EquivCNP ($E$) | $0.2213 \pm 0.0007$ |
| EquivCNP ($\widetilde{E}$) | $0.1992 \pm 0.0010$ |

**Log-likelihood of context data** We assess the model's ability to reconstruct the context set by setting the target set equal to the context set and computing the log-likelihood. Note that, unlike the log-likelihood values from Table 1, where the target range expands beyond the context range, now the model is just tested within the context range, leading to higher overall values.

The results in Table 3 show that the model is able to accurately model the context sets, with log-likelihood values close to the ground truth log-likelihood of the data of $0.2806 \pm 0.0005$.

**Analysis of equivariance deviation** We next analyse the approximately equivariant behaviour of our models. More specifically, we show that while breaking equivariance to a certain degree, the models still share similarities with the fully-equivariant models—thus being characterised as approximately equivariant. This is already visually depicted in Figure 1, where the predictive means of the approximately equivariant models are similar to the predictive means of the corresponding equivariant models up to some 'small perturbation'.

We also show this quantitatively, by introducing the equivariance deviation

$$\Delta_{\text{equiv}} = \mathbb{E}\left[\left\| \frac{\mu_{\text{equiv}} - \mu_{\text{approx-equiv}}}{\mu_{\text{equiv}}} \right\|_p\right],$$

where $\mu_{\text{approx-equiv}}$ represents the predictive mean of the approximately equivariant model, and $\mu_{\text{equiv}}$ the predictive mean of the same model with zeroed-out basis functions (i.e. the equivariant component of the approximately equivariant model). In this case, we consider the $L_1$ norm (i.e. $p = 1$).

**Table 4:** Equivariance deviation ($\Delta_{\text{equiv}}$) of the approximately equivariant models in the 1-D synthetic GP experiment.

| Model | $\Delta_{\text{equiv}}$ |
|---|---|
| TNP ($\widetilde{T}$) | $0.0896 \pm 0.0011$ |
| ConvCNP ($\widetilde{T}$) | $0.0823 \pm 0.0005$ |
| RelaxedConvCNP ($\widetilde{T}$) | $0.1460 \pm 0.0006$ |
| EquivCNP ($\widetilde{E}$) | $0.0825 \pm 0.0006$ |

The equivariance deviations from Table 4 are around $8 - 9\%$ (with the exception of the Relaxed-ConvCNP ($\widetilde{T}$) which shows a $15\%$ deviation). This indicates that the approximately equivariant models' predictions deviate only slightly from the equivariant predictions, thus being equivariant in the approximate sense. Moreover, as pictured in Figure 1, the deviation from strict equivariance is not random, and allows the models to learn symmetry-breaking features in a data-driven manner (in this case, the lengthscale changepoint).

**Analysis of number of fixed inputs** In this ablation, we analyse the effect of the number of fixed inputs $B$. As mentioned in the main text, the number of fixed inputs influences the number of degrees of freedom in which the decoder deviates from $G$-equivariance. Thus, 0 fixed inputs leads to a

$G$-equivariant model, while in the limit of infinitely many fixed inputs, the decoder can become fully non-equivariant.

In the main experiment we used $64$ fixed inputs for the ConvCNP ($\widetilde{T}$) and $4$ Fourier coefficients for TNP ($\widetilde{T}$). Table 5 shows the results when varying the number of fixed inputs $B$ for the two models. We observe that the biggest gain in performance is achieved by going from $B = 0$ to $B = 1$ basis functions. In this dataset, the approximate equivariance is achieved by modifying one parameter (i.e. lengthscale) of the data generation process from one side of the changepoint location to the other. Indeed, for the TNP ($\widetilde{T}$) we observe that the performance plateaus for $B \geq 1$, suggesting that one Fourier coefficient is enough to capture the symmetry-breaking feature of the dataset. For the ConvCNP ($\widetilde{T}$) the performance increases up to $B \geq 4$, and for $B \geq 1$ the test log-likelihoods are within 3 standard deviations for all variants. Thus, the empirical findings are in agreement with the data generation process, indicating that deviation in one or a couple of degrees of freedom suffices to capture the departure from strict equivariance. What is more, we see that using more fixed inputs than necessary does not hurt the performance of the model (when considering the standard deviations). This motivates in practice the use a 'sufficiently high number' of fixed inputs, to make sure the model is given enough flexibility to correctly capture the departure from strict equivariance.

**Table 5:** Average test log-likelihoods ($\uparrow$) of the TNP ($\widetilde{T}$) and the ConvCNP ($\widetilde{T}$) models when varying the number of fixed inputs. All standard deviations are 0.004.

| | No. fixed inputs | | | | | |
| --- | --- | --- | --- | --- | --- | --- |
| Model | 0 | 1 | 2 | 4 | 8 | 16 |
| TNP ($\widetilde{T}$) | $-0.488$ | $-0.409$ | $-0.410$ | $-0.406$ | $-0.408$ | $-0.403$ |
| ConvCNP ($\widetilde{T}$) | $-0.499$ | $-0.451$ | $-0.439$ | $-0.430$ | $-0.433$ | $-0.432$ |

### E.2 Smoke Plumes

The smoke plume dataset consists of $128 \times 128$ 2-D smoke simulations for different initial conditions generated through PhiFlow[Holl et al., 2020]. Hot smoke is emitted from a circular region at the bottom, and the simulations output the resulting air flow in a closed box. We also introduce a fixed obstacle at the top of the box. To obtain a variety of initial conditions we sample the radius of the smoke source uniformly according to $r \sim \mathcal{U}[5, 30]$. Moreover, we randomly choose its position among three possible x-axis locations: $\{30, 64, 110\}$ (but we keep the $y$ position fixed at 5). Finally, we sample the buoyancy coefficient of the medium according to $B \sim \mathcal{U}[0.1, 0.5]$. The closed box, the fixed spherical obstacle, and the position of the smoke inflow (sampled out of three possible locations) break the symmetry in this dynamical system.

For each initial condition, we run the simulation for 35 time-steps with a time discretisation of $\Delta t = 0.5$, and only keep the last state as one datapoint. We show in Figure 5 examples of such states. In total, we generate samples for 25,000 initial condition, and we use 20,000 for training, 2,500 for validation, and the remaining for test.

The inputs consist of $[32, 32]$ regions sub-sampled from the $[128, 128]$ grid. Each dataset consist of a maximum of $N = 1024$ datapoints, from which the number of context points are sampled according to $N_c \sim U\{10, 250\}$, with the remaining points set as the target points.

We use an embedding / token size of $D_z = 128$ for the PT-TNP-based models and $D_z = 16$ for the ConvCNP-based models. The decoder consists of an MLP with two hidden layers of dimension $D_z$. The decoder parameterises the mean and pre-softplus standard deviation of a Gaussian likelihood with heterogeneous noise. Model specific architectures are as follows: **PT-TNP** For the PT-TNP models we use the same architecture dimensions as the TNP described in Appendix E.1. We use the IST-style implementation of the PT-TNP [Ashman et al., 2024], with initial pseudo-token values sampled from a standard normal distribution. We use 128 pseudo-tokens.

**PT-TNP** ($T$) The PT-TNP ($T$) models adopt the same architecture choices as the TNP ($T$) described in Appendix E.1. The initial pseudo-tokens and pseudo-input-locations are sampled from a standard

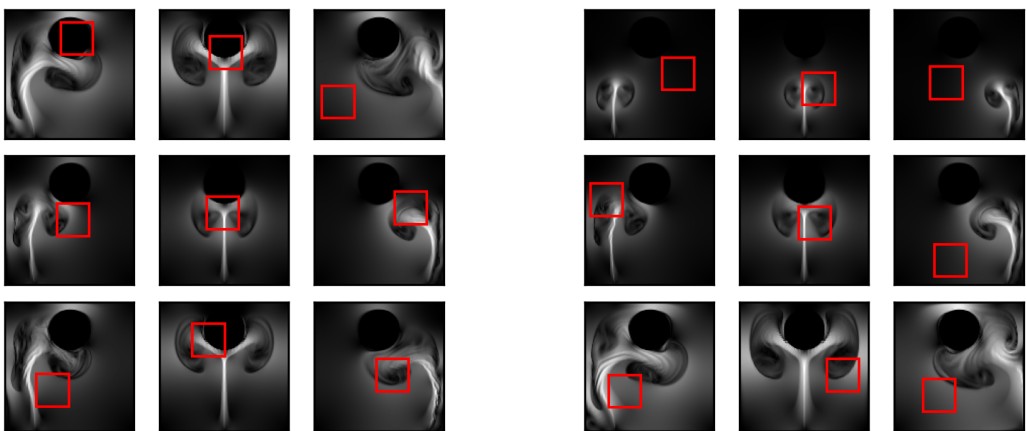

**Figure 5:** Examples of smoke simulations from the smoke plume dataset for six different combinations of smoke radius $r$ and buoyancy $B$. For each such combination, we show the resulting state for all of the three possible x-axis locations. The inputs to our models are randomly sampled $32 \times 32$ patches (indicated in red) from the $128 \times 128$ states.

normal. We use 128 pseudo-tokens. Pseudo-code for a forward pass through the PT-TNP ($T$) can be found in Appendix B.4.

**PT-TNP ($\widetilde{T}$)** The architecture of the PT-TNP ($\widetilde{T}$) is similar to that of the PT-TNP ($T$), with the exception of the extra fixed inputs that are added to the pseudo-token representation. These are obtained by passing the pseudo-token locations through an MLP with two hidden layers of dimension $D_z$. We use $D_z$ fixed inputs and apply dropout to them with a probability of $0.5$. Pseudo-code for a forward pass through the PT-TNP ($\widetilde{T}$) can be found in Appendix B.4.

**ConvCNP ($T$)** For the ConvCNP model, we use a U-Net [Ronneberger et al., 2015] architecture for the CNN with 9 layers. We use $C_{in} = 16$ input channels and $C_{out} = 16$ output channels, with the number of channels doubling / halving on the way down / up. Between each downwards layer we apply pooling with size two, and between each upwards layer we linearly up-sample to recover the size. We use a kernel size of $k = 9$ with a stride of one. We use the natural discretisation of the $128 \times 128$ grid.

**ConvCNP ($\widetilde{T}$)** We use the same architecture as the ConvCNP ($T$). The fixed inputs are obtained by passing the discretised grid locations through an MLP with two hidden layers of dimension $C_{in}$. We use $C_{in}$ fixed inputs and apply dropout with probability $0.5$.

**RelaxedConvCNP ($\widetilde{T}$)** We use an identical architecture to the ConvCNP ($T$), with regular convolutional operations replaced by relaxed convolutions. We use $L = 1$ kernels such that the total parameter count remains similar. We obtain the additional fixed inputs by passing the effective discretised grid at each layer (after applying the same pooling / up-sampling operations as the U-Net) through an MLP with two hidden layers of dimension $C_{in}$.

**EquivCNP ($E$)** For the EquivCNP ($E$) model, we use a steerable $E$-equivariant CNN architecture consisting of nine layers, each with $C = 16$ input / output channels. We use a kernel size of $k = 9$ and stride of one. We discretise the continuous rotational symmetries to integer multiples of $2\pi/8$. We use the natural discretisation of the $128 \times 128$ grid.

**EquivCNP ($\widetilde{E}$)** We use the same architecture as the EquivCNP ($E$) with the same fixed input architecture as the ConvCNP ($\widetilde{T}$).

**Training Details and Compute** For all models, we optimise the model parameters using AdamW [Loshchilov and Hutter, 2017] with a learning rate of $5 \times 10^{-4}$. For the ConvCNP $T$ and $\widetilde{T}$, EquivCNP $T$ and $\widetilde{T}$, and non-equivariant PT-TNP models we use a batch size of 16, while for the PT-TNP $T$ and $\widetilde{T}$ models we use a batch size of 8. Gradient value magnitudes are clipped at $0.5$. We train for a maximum of 500 epochs, with each epoch consisting of 16,000 datasets for a batch size of 16, and

8,000 datasets for a batch size of 8 (10,000 iterations per epoch). We evaluate the performance of each model on 80,000 test datasets. We train and evaluate all models on a single 11 GB NVIDIA GeForce RTX 2080 Ti GPU.

**Additional Results** We show in Figure 6 a comparison between the predictive means, as well as the absolute difference between them and the ground-truth (GT), for all the models in Table 1. For PT-TNP and ConvCNP, the predictions of the equivariant models are more blurry, whereas the approximately equivariant models better capture the detail surrounding the obstacle or the flow boundary. For the EquivCNP we did not observe a significant difference between the equivariant and approximately equivariant model.

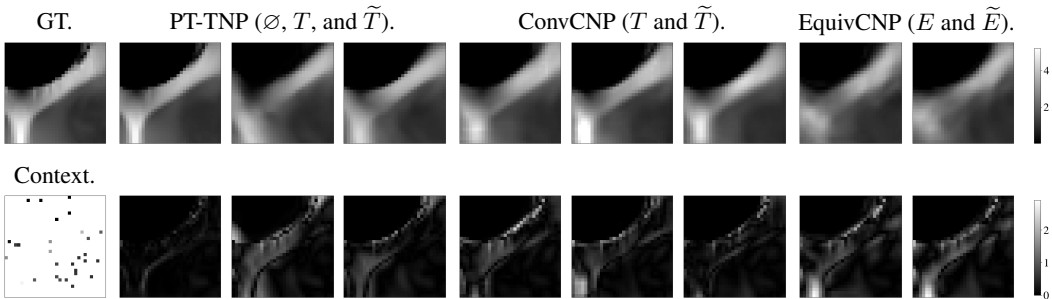

**Figure 6:** A comparison between the predictive distributions of the equivariant and approximately equivariant versions of the three classes of models: PT-TNP, ConvCNP, and EquivCNP. From left to right we show: the ground-truth (GT), the non-equivariant PT-TNP, PT-TNP ($T$), PT-TNP ($\widetilde{T}$), ConvCNP ($T$), ConvCNP ($\widetilde{T}$), RelaxedConvCNP ($\widetilde{T}$), EquivCNP ($E$), and EquivCNP ($\widetilde{E}$). The top row shows the mean of the predictions, while the bottom row shows the absolute difference between the predicted mean of each model and the ground-truth.

### E.3 Environmental Data

The environmental dataset consists of surface air temperatures derived from the fifth generation of the European Centre for Medium-Range Weather Forecasts (ECMWF) atmospheric reanalyses (ERA5) [Copernicus Climate Change Service, 2020]. The data has a latitudinal and longitudinal resolution of $0.5°$, and temporal resolution of an hour. We consider data collected in 2018 and 2019 from regions in Europe (latitude / longitude range of $[35°, 60°]$ / $[10°, 45°]$) and in the US (latitude / longitude range of $[-120°, -80°]$ / $[30°, 50°]$). In both the 2-D and 4-D experiment, we train on Europe's 2018 data and test on both Europe's and the US's 2019 data.

For the 2-D experiment, the inputs consist of latitude and longitude values. Individual datasets are obtained by sub-sampling the larger regions, with each dataset consisting of a $[32, 32]$ grid spanning $16°$ across each axis. For the 4-D experiment, the inputs consist of latitude and longitude values, as well as time and surface elevation. Each dataset consists of a $[4, 16, 16]$ grid spanning $8°$ across each axis and 4 days. In both experiments, each dataset consists of a maximum of $N = 1024$ datapoints, from which the number of context points are sampled according to $N_c \sim \mathcal{U}\{\lfloor \frac{N}{100} \rfloor, \lfloor \frac{N}{3} \rfloor\}$, with the remaining set as target points.

For all models, we use a decoder consisting of an MLP with two hidden layers of dimension $D_z$. The decoder parameterises the mean and pre-softplus variance of a Gaussian likelihood with heterogeneous noise. Model specific architectures are as follows:

**PT-TNP** Same as Appendix E.2.

**PT-TNP** ($T$) Same as Appendix E.2.

**PT-TNP** ($\widetilde{T}$) Same as Appendix E.2 with 128 pseudo-tokens and fixed inputs zeroed outside 2018 and the latitude / longitude range of $[35°, 60°]$ / $[10°, 45°]$.

**ConvCNP** ($T$) Same as Appendix E.2.

**ConvCNP** ($\widetilde{T}$) Same as Appendix E.2 with fixed inputs zeroed outside 2018 and the latitude / longitude range of $[35°, 60°]$ / $[10°, 45°]$.

**RelaxedConvCNP** ($\widetilde{T}$) Same as Appendix E.2 with fixed inputs zeroed outside 2018 and the latitude / longitude range of $[35°, \, 60°] \, / \, [10°, \, 45°]$.

**EquivCNP** ($E$) Same as Appendix E.2.

**EquivCNP** ($\widetilde{E}$) Same as Appendix E.2 with fixed inputs zeroed outside 2018 and the latitude / longitude range of $[35°, \, 60°] \, / \, [10°, \, 45°]$.

**Training Details and Compute** For all models, we optimise the model parameters using AdamW [Loshchilov and Hutter, 2017] with a learning rate of $5 \times 10^{-4}$ and batch size of 16. Gradient value magnitudes are clipped at 0.5. We train for a maximum of 500 epochs, with each epoch consisting of 16,000 datasets (10,000 iterations per epoch). We evaluate the performance of each model on 16,000 test datasets. We train and evaluate all models on a single 11 GB NVIDIA GeForce RTX 2080 Ti GPU.

**Additional Results** To demonstrate the importance of dropping out the fixed inputs during training with finite probability, we compare the performance of two ConvCNP ($\widetilde{T}$) models on the 2-D experiment in Table 6: one with a dropout probability of $0.0$, and the other with $0.5$. Due to limited time, we were only able to train each model for 300 epochs (rather than for 500 epochs, hence the difference in results for the ConvCNP ($\widetilde{T}$) model with dropout probability of $p = 0.5$ to those shown in Table 2). Nonetheless, we observe that having a finite dropout probability is important for the model to be able to generalise OOD (i.e. the US).

**Table 6:** Average test log-likelihoods ($\uparrow$) for the 2-D environmental regression experiment. $p$ denotes the probability of dropping out the fixed inputs during training.

| Model | Europe ($\uparrow$) | US ($\uparrow$) |
|---|---|---|
| ConvCNP ($\widetilde{T}, p = 0.0$) | $1.19 \pm 0.01$ | $-0.51 \pm 0.02$ |
| ConvCNP ($\widetilde{T}, p = 0.5$) | $1.16 \pm 0.01$ | $0.16 \pm 0.02$ |

**Log-likelihood of context data** We perform the same assessment as in Appendix E.1 to check whether the model is able to accurately reconstruct the context data.

The results in Table 7 show that the models have good performance when tested on the context sets, achieving better log-likelihoods than in Table 2, where the models are tested on target sets.

**Analysis of equivariance deviation** We repeat the equivariance deviation analysis from Appendix E.1 on the 2-D environmental regression dataset. In this case we use the $L_2$ instead of the $L_1$ norm. Note that we only compute this on Europe, since the predictions on US are obtained by zeroing-out the basis function, leading to an equivariance deivation of $0$. The results are shown in Table 8.

All equivariance deivations are between $2 - 4\%$, indicating that the models only slightly deviate from the equivariant prediction. However, as shown in Figure 2, this deviation allows them to capture the

**Table 7:** Average test log-likelihoods ($\uparrow$) for the 2-D environmental regression experiment when tested on context sets.

| | 2-D Regression | |
|---|---|---|
| Model | Europe ($\uparrow$) | US ($\uparrow$) |
| PT-TNP | $1.74 \pm 0.01$ | $-$ |
| PT-TNP ($T$) | $1.66 \pm 0.01$ | $1.28 \pm 0.01$ |
| PT-TNP ($\widetilde{T}$) | $1.76 \pm 0.01$ | $1.47 \pm 0.01$ |
| ConvCNP ($T$) | $1.20 \pm 0.02$ | $0.34 \pm 0.02$ |
| ConvCNP ($\widetilde{T}$) | $1.50 \pm 0.01$ | $0.97 \pm 0.01$ |
| RelaxedConvCNP ($\widetilde{T}$) | $1.29 \pm 0.01$ | $0.86 \pm 0.01$ |
| EquivCNP ($E$) | $2.03 \pm 0.01$ | $1.76 \pm 0.01$ |
| EquivCNP ($\widetilde{E}$) | $2.05 \pm 0.01$ | $1.69 \pm 0.01$ |

**Table 8:** Equivariance deviation ($\Delta_{\text{equiv}}$) of the approximately equivariant models in the 2-D environmental regression experiment.

| Model | $\Delta_{\text{equiv}}$ |
|---|---|
| PT-TNP ($\widetilde{T}$) | $0.0406 \pm 0.0005$ |
| ConvCNP ($\widetilde{T}$) | $0.00237 \pm 0.0004$ |
| RelaxedConvCNP ($\widetilde{T}$) | $0.0239 \pm 0.0004$ |
| EquivCNP ($\widetilde{E}$) | $0.0242 \pm 0.0005$ |

symmetry-breaking components of the weather dataset, with one important such component being the topography.

**Analysis of number of fixed inputs** Finally, we investigate the influence of the number of fixed inputs $B$ on the performance of the model. We vary the number of fixed inputs used in the ConvCNP ($\widetilde{T}$) from the 2-D regression experiment. The fixed inputs are linearly transformed to 16 inputs, which are then summed together with the input into the CNN.

The results are shown in Table 9, where we see that, similarly to the 1-D GP experiment, the largest gain in performance is obtained by going from 0 to a single fixed input. Using more than 1 fixed input provides diminishing gains, but, importantly, never hurts performance. The saturation in performance for $B \geq 1$ is expected in this dataset, given that the information that is primarily missing is the topography. However, in practice, given that the performance of the model does not decrease with increasing $B$ (i.e. the model learns to ignore unnecessary fixed inputs), one can choose a 'sufficiently high' value for $B$.

**Table 9:** Average test log-likelihoods ($\uparrow$) of the ConvCNP ($\widetilde{T}$) with different number of additional fixed inputs for the 2-D environmental regression experiment. We show the results when the models are tested on Europe.

| Region | No. fixed inputs | | | | | |
|---|---|---|---|---|---|---|
| | 0 | 1 | 2 | 4 | 8 | 16 |
| Europe | $1.11 \pm 0.01$ | $1.19 \pm 0.01$ | $1.19 \pm 0.01$ | $1.20 \pm 0.01$ | $1.20 \pm 0.01$ | $1.20 \pm 0.01$ |

