# OpenReview forum: "Approximately Equivariant Neural Processes"
_NeurIPS.cc/2024/Conference — NeurIPS 2024 poster_

### Official Review · Reviewer_dBf7 · 2024-07-08

**Soundness:** 2
**Presentation:** 1
**Contribution:** 2
**Rating:** 5
**Confidence:** 4

**Summary:**

This paper implemented the approximately equivariant neural process (NP) by relaxing the equivariance of the NP decoder and the relaxation is conducted by adding several leanable parameters as fixed inputs fed with the data embeddings into the decoder.

**Strengths:**

The proposed method is applicable regardless of architecture and easy to be implemented because it is simply adding some parameters as inputs.

**Weaknesses:**

1. No supplemental material for a source code.
1. A concept figure will be helpful to understand the intuition of the paper.
1. Absent analysis for equivariance error of the trained models. Did the trained model really reflect the approximate equivariance inherited in the dataset?
1. The paper reported only the log-likelihood of (maybe) the target data, which may be overfitted by the target data and underfitted by the context data.  It is necessary to report the log-likelihoods with respect to both context and target separately to show the strength of the suggested method accurately.
1. Analysis for the number of the fixed inputs is necessary in order to prove that the number of the fixed input really controls the magnitude of the approximate equivariance.
1. All included experiments are only regression tasks. Since NP is essentially a meta-learner, it would be better to add experiments beyond regression to demonstrate its generalization performance across various types of tasks.

**Questions:**

1. Encoder should be equivariant? If so, doesn’t it lose non-equivariant information from the dataset D?
1. How to add the fixed inputs into the dataset embedding in ConvCNP? If we just sum all, the number of the fixed inputs is obviously meaningless.
1. When the decoder is ViT, how many model parameters are additionally necessary compared to the strictly equivariant case?
1. It seems that the suggested relaxation is applicable to not only NP. Why did the paper choose NP to prove the relaxation idea?

**Limitations:**

There is no limitation of the contribution itself but experimental limitations are mentioned in the weakness.

---

> ### Author Rebuttal · Authors · 2024-08-07
>
> Thank you for your feedback and for recognising how easily our method can be applied to achieve approximate equivariance.
>
> **Supplemental code** - As mentioned in the Paper Checklist, we intend to provide open access to the data and code prior to publication.
>
> **Concept figure** - Please see the PDF attached in the common response for a sketch concept figure. If you find this figure helpful, we will include a refined version in the revised paper.
>
> **Equivariance error** - We believe we do provide qualitative analysis of the model’s behaviour with regards to equivariance. For example, in the GP experiment, we provide both the model predictions, as well as the predictions of the model if we set the additional fixed inputs to zero (corresponding to the equivariant component of the model). Fig. 1 e), f), g), and h) show how the additional fixed inputs help the models better capture the transition from the low-lengthscale to the high-lengthscale region (also see lines 290-293).
>
> Another example is provided in the environmental data experiment. One significant variable influencing the surface air temperature is the elevation. If the elevation map is not provided to the model, we expect the model to depart from strict translation equivariance in a way that correlates with the elevation. This is exactly what we test in the 2-D Spatial Regression task, where we see that the model can infer the effect of local topographical features on air temperature through the non-equivariant component. This can be observed by comparing the difference between the approximately equivariant predictions and the equivariant component of the model (Fig. 2f, i) with the true elevation map (Fig. 2c). Remarkably, the non-equivariant component (i.e. difference between predictions and equivariant component) learns a representation that closely resembles the elevation map, which is probably one of the main factors that leads to departure from strict equivariance in this setup.
>
> **Log-likelihood of target data** - In the usual meta-learning framework for neural processes, the goal is to predict the target set given the context set, so models only parametrise the probability of the target set given the context. For example, see [1] and many follow-up publications [2, 3, 4, 5, 6].
>
> [1] Garnelo, M. et al. (2018). Conditional Neural Processes.
>
> [2] Garnelo, M. et al. (2018). Neural Processes.
>
> [3] Kim, H. et al. (2019). Attentive Neural Processes.
>
> [4] Gordon, J. et al. (2019). Convolutional Conditional Neural Processes.
>
> [5] Bruinsma, W.P. et al. (2023). Autoregressive Conditional Neural Processes.
>
> [6] Nguyen, T., & Grover, A. (2022). Transformer Neural Processes: Uncertainty-Aware Meta Learning Via Sequence Modeling.
>
> **Analysis of number of fixed inputs** - Please refer to our common response.
>
> **Limited to regression** - NPs can indeed be used for other types of tasks beyond regression, like classification. The focus of our submission is on approximate equivariance, and we are primarily interested in applications in regression for studying this property (e.g. fluid dynamics simulation and environmental data modelling). This is also the approach that other related work studying approximate equivariance take, focussing primarily on regression problems [1]. Nevertheless, we acknowledge that applications in classification could also be investigated, similarly to [2, 3], but this is beyond the intended scope of our work.
>
> [1] Wang, R. et al. (2022). Approximately Equivariant Networks for Imperfectly Symmetric Dynamics.
>
> [2] Elsayed, G. F. et al. Revisiting spatial invariance with low-rank local connectivity. ICML 2020.
>
> [3] van der Ouderaa T. et al. Relaxing equivariance constraints with non-stationary continuous filters. NeurIPS 2022.
>
> **Equivariant encoder** - Encoders of equivariant NP architectures are indeed equivariant. Importantly, these encoders do not lose information, because they are usually so-called homeomorphisms, which means that they map every different dataset to a different encoding ([1] , App.A of [2]). Equivariance then means, for example, that the encoding is shifted whenever the dataset is shifted. Equivariance does not break bijectivity (i.e. does not lose information), but says something about how the encoding behaves as the dataset is transformed by a symmetry.
>
> [1] Zaheer M. et. al (2017). Deep Sets. NIPS'17.
>
> [2] Gordon, J. et. al. (2019). Convolutional Conditional Neural Processes. ICLR 2020.
>
> **Adding fixed inputs into the dataset embedding** - We provide implementation details in Appendix D.  When adding the fixed inputs to the dataset embedding for the ConvCNP, we achieve this by adding a different fixed input to each input channel into the CNN. This isn’t the only route through which additional fixed inputs can be included—if we were to treat them as additional channels into the CNN, then the number of additional fixed inputs is not restricted.
>
> **ViT decoder** - In general, the only additional parameters required to modify existing equivariant architectures are those that are required to define the additional fixed inputs. This is typically much smaller than the number of parameters used in the original architecture. For example, for the 2-D environmental data experiment, the regular ConvCNP ($T$) has $15.886 \times 10^6$ parameters, whereas the ConvCNP ($\widetilde{T}$) has $15.887\times 10^6$ parameters (an increase of < 0.01%). The regular TNP ($T$) has $4.006 \times 10^4$ parameters, whereas the TNP ($\widetilde{T}$) has $4.823 \times 10^4$ parameters (an increase of around 20%).
>
> **Focus on NPs** - Please see response to reviewer WhxG regarding **Focus on NPs**.
>
> We appreciate that our analysis combines results from the neural process literature with more formal mathematical analysis, and consequently might be more dense than usual. Nevertheless, we hope that you find our rebuttal informative, and we would appreciate it if you would consider increasing your score.

---

> ### Comment · Reviewer_dBf7 · 2024-08-10
> **Rebuttal**
>
> Thank you so much for the detailed clarification. I understood much of it, but I still have some concerns regarding the log-likelihood and the equivariance error.
>
> **The log-likelihood of reconstruction for the context data is important to evaluate the practical performance of neural processes.**
>
> For example, we often set the variance of the output as $0.1+0.9\cdot\text{Softplus}(\sigma)$ where $\sigma$ is the decoder output for variance. However, the Transformer Neural Process (TNP) sets it as $\exp(\sigma)$, which differed from its baselines. This setting results in high variance that can cover lots of the target data but remains high even for the context data, leading to significant underfitting in the context data. Indeed, TNP shows a log-likelihood in context reconstruction that is 10 times worse than its baselines. A good likelihood only in the target data may still seem favorable because its variance easily covers the true target data, but it simply maintain high uncertainty for all data, which is not practical as a meta-learner. For these reasons, many previous works also report the log-likelihood or error of context data reconstruction. Please refer to Figure 3 in ANP [1], Table 1 in BNP [2], and Table 1 in MPNP [3].
>
> If you agree with my opinion, I hope you consider including the log-likelihood or error of context data reconstruction, which is not difficult to measure.
>
> [1] Kim, H. et al. (2019), Attentive Neural Processes.
>
> [2] Lee, J. et al. (2020), Bootstrapping Neural Processes.
>
> [3] Lee, H. et al. (2023), Martingale Posterior Neural Processes.
>
> **Equivariance error analysis, either in theory or experiment, is necessary.**
>
> What I mean by equivariance error analysis is, for example, adopting a simple function like $y=\text{CosineSimilarity}(\boldsymbol{x}\_1,\boldsymbol{x}\_2)$ (rotation-invariant), then weakly breaking the symmetry, such as by using $y = \text{CosineSimilarity}(\boldsymbol{x}\_1, \boldsymbol{x}\_2+0.01\cdot\boldsymbol{1})$, and generating a synthetic dataset. Note that since we already know the true function and the actual equivariance error, we can compare the equivariance error learned from the synthetic dataset with the true error.
>
> The paper theoretically showed that adding fixed learnable inputs breaks the equivariance but it does not showed that how much it breaks and whether it can indeed utilize the inductive bias of the weakly broken symmetry or not. If you show such analysis, you can say that the reason why your method works well is not because the method fully break the inductive bias and just empower the expressivity of the neural network. Please refer to Figure 3 in RPP [1] and Figure 3 in PER [2].
>
> [1] Finzi, M. et al. (2021), Residual Pathway Priors for Soft Equivariance Constraints.
>
> [2] Kim, H. et al. (2023), Regularizing Towards Soft Equivariance Under Mixed Symmetries.

---

> > ### Author Response · Authors · 2024-08-11
> > **Answer Part 1**
> >
> > Thank you for taking the time to go through our rebuttal. We address your remaining concerns below.
> >
> >
> > **Log-likelihood of context data.**
> > The practical importance of neural processes is determined by how they are used in practice, after training. In the usual framework, after training, there will be an unseen context set $D^*_c$ and an unseen target set $D^*_t$, and practical utility of the neural process is determined by how well the model predicts $D^*_t$ given $D^*_c$, usually measured in terms of log-likelihood. The appropriate way of measuring this performance is to (a) hold out some context–target pairs from training; and (b), after training, to compute the log-likelihood of the held-out target sets given the context sets.
> >
> >
> > In this framework, the probabilities $p(D_c | D_c)$ can be reasonably used as a diagnostic, but these probabilities do not necessarily correlate with good performance on the task of interest (predicting $D^*_t$ given $D^*_c$). For example, consider a NP perfectly trained on data sampled from a GP with a Matern kernel, and suppose $(D^*_c, D^*_t)$ is now sampled from a GP with a periodic kernel. Then the NP will achieve near-optimal $p(D^*_c | D^*_c)$, because the NP can reconstruct data it is conditioned on; but the NP will achieve disastrous $p(D^*_t | D^*_c)$, because the NP has never interpolated periodic data before.
> >
> >
> > It is often the case that the sampling distributions of the context and target data are the same. In that case, a model that maximises $\log p(D_t | D_c)$ will be guaranteed to reconstruct the context data well, because a context set is like a target set.
> >
> >
> > **“A good likelihood only in the target data may still seem favorable because its variance easily covers the true target data”**
> >
> > Because probability densities integrate to one, the likelihood will penalise both too high and too low variance. In fact, the optimal likelihood is achieved if and only if the variance is equal to the true variance and the mean equal to the true mean. The likelihood therefore cannot be “cheated” by artificially increasing the variance. To make this mathematically rigorous, see e.g. Prop 1 by Foong et al., 2020.
> >
> >
> > In summary, the right way to measure practical performance is by computing log-likelihoods of held-out target data given context data, and this metric cannot be “cheated” by artificially increasing the variance. Nevertheless, we agree that the log-likelihood for reconstructing the context set can be used as a diagnostic metric.
> >
> >
> > If you insist that this diagnostic metric is important, then we are willing to include it as an additional result in the supplementary material.
> >
> > Finally, as a clarification regarding implementation - **“(...) we often set the variance of the output as 0.1+0.9⋅Softplus($\sigma$), where $\sigma$ is the decoder output for variance. However, the Transformer Neural Process (TNP) sets it as $\exp⁡(\sigma)$”**
> >
> > Our implementation of the TNP parameterises the standard deviation of the normal distribution as $\sigma_{min} + \text{Softplus}(z)^{1/2}$, where $z$ is the output of the decoder for the variance and $\sigma_{min}$ a hyperparameter usually set at 0.1. This is mentioned in the Appendix lines 700-701: “The decoder parameterises the mean and pre-softplus variance [correction: standard deviation] of a Gaussian likelihood with heterogeneous noise.”
> >
> > **References**
> >
> > Foong, Andrew Y. K. et al. Meta-Learning Stationary Stochastic Process Prediction with Convolutional Neural Processes. NeurIPS 2020.

---

> > ### Author Response · Authors · 2024-08-11
> > **Answer Part 2**
> >
> > **Equivariance error analysis.**
> >
> > **“(...) it does not showed that how much it breaks and whether it can indeed utilize the inductive bias of the weakly broken symmetry or not.”**
> >
> > In the experiments, we compare the approximately equivariant model to the strictly equivariant model obtained by setting the additional inputs to zero. This reveals exactly what the effect of the additional inputs is and in what way they break equivariance. In particular, the bottom row of Figure 1 shows predictions for both the approximately equivariant model and the equivariant model obtained by setting the additional inputs to zero. Note that the equivariant model unnecessarily inflates the uncertainty, because translation equivariance prevents the model from recognising a fixed $x$-location, so it cannot learn that a change-point always happens at $x=0$.
> >
> >
> > The same comparison is done in Figure 2 for the environmental data experiment, (d) versus (e) and (g) versus (h). In the same figure, (f) and (i) show how much the predictions change as a consequence of the additional inputs. This difference exactly correlates with the missing information, the orography in (c).
> >
> >
> > We believe that these comparisons show that the additional inputs break equivariance to an appropriate extent. For example, in the bottom row of Figure 1, note that the approximately equivariant and equivariant predictions are very similar before the length-scale change. Including the additional inputs only changes the predictions after the length-scale change.
> >
> > **“If you show such analysis, you can say that the reason why your method works well is not because the method fully break the inductive bias and just empower the expressivity of the neural network.”**
> >
> > The additional basis functions do strictly increase the expressivity of the model, exactly because the basis functions break equivariance of the model. That is, if the basis functions are zero, then the model is fully equivariant; and if the basis functions are non-zero, then the model is not equivariant. Because everything is continuous, the transition between these two “modes” is smooth and depends on the magnitude of the basis functions, which the model automatically learns.
> >
> > The main practical question that remains is to whether this indeed corresponds to useful, practical, and appropriate improvements. We believe that is clearly shown in Figures 1 and 2.
> >
> > We agree that it would be very interesting to more rigorously quantify the degree to which equivariance is broken with a metric like in RPP. However, for the current submission, we believe to have demonstrated that additional inputs (a) have a principled theoretical motivation, (b) break equivariance in interpretable and appropriate ways (as argued above), and (c) give considerable performance improvements on real-world data (experimental results). Although such an analysis would be interesting, we believe to have shown that additional inputs are a practical method that works in the intended way and have therefore left it for future work.
> >
> > Nevertheless, if you insist that a more rigorous quantification is essential, then we would be willing to compute a metric very similar to EquivError from RPP for Figures 1 and Figure 2. For example, for Figure 2, this would divide the norm for panel (f) by the average of the norms of panels (d) and (e). By the colour bars, we expect this value to be around 2%. We would include these numbers in the captions.

---

> ### Comment · Reviewer_dBf7 · 2024-08-12
>
> Thank you for providing more information, but most of it seems to carry the same nuance as the first rebuttal.
>
> **Context Log-Likelihood**
>
> As you mentioned, the context log-likelihood is not a direct performance metric. However, high uncertainty in the reconstruction of context indicates merely inferring the whole sequence from a pattern in the context, which is not a stochastic process. Instead, it is more like regression using a neural network that takes the context as input and produces the target as output. For example, in a task like prediction from irregularly sampled time series, the reason we use a stochastic process rather than other frameworks is to analyze the statistics of the time series, including both the context and target data, especially since the context itself may contain observational noise. If the goal were simply to build a model that reconstructs the target well, I would prefer using imputation models over neural processes. Target performance is not easily manipulated by variance, but it can be artificially enhanced by sacrificing context performance, as seen in TNP.
>
> **Equivariance Error**
>
> I understand that you showed the approximately equivariant model outperformed the equivariant model (with added inputs set to zero). However, my point is more about comparing the fully non-equivariant model with the approximately equivariant model. The approximately equivariant model should outperform the fully non-equivariant model because it leverages the inductive bias. While you did make this comparison, it does not guarantee that your model is indeed an approximately **equivariant** model, as you did not demonstrate that it truly reflects the actual approximate equivariance. I think your paper focused more on how to create an approximately **equivariant** model, rather than just breaking equivariance to overcome the strict constraints of equivariant architectures and add flexibility. If your paper is solely about breaking equivariance to introduce flexibility, then the experiments I suggested might not be necessary.
>
> To be honest, I do not understand why you do not report the log-likelihoods during rebuttal period, especially since you agree that that is also a diagnostic metric. In my experience, **measuring the context log-likelihood takes less than 10 minutes**.
>
> I think the paper should be reviewed after including the context log-likelihoods and the equivariance errors.

---

> > ### Author Response · Authors · 2024-08-13
> > **No comment notification**
> >
> > We have not received any notification after posting the previous two comments. With this comment, we just wanted to make sure that the reviewer gets notified about the new comments.

---

> > ### Author Response · Authors · 2024-08-13
> >
> > Here are the additional results for the environmental data.
> >
> > **Context log-likelihoods environmental data**
> >
> > Table 1: Average context loglikelihoods for the 2-D environmental regression experiment. Results are grouped together by model class.
> > _________________________________
> > Model | Europe | US
> > __________________
> > PT-TNP | 1.74 $\pm$ 0.01 | - |
> >
> > PT-TNP ($T$) | 1.66 $\pm$ 0.01 | 1.28 $\pm$ 0.01
> >
> > PT-TNP ($\widetilde{T}$) | 1.76 $\pm$ 0.01 | 1.47 $\pm$ 0.01
> > ______________________________________
> > ConvCNP ($T$) | 1.20 $\pm$ 0.02 | 0.34 $\pm$ 0.02
> >
> > ConvCNP ($\widetilde{T}$) | 1.50 $\pm$ 0.01 | 0.97 $\pm$ 0.01
> >
> > RelaxedConvCNP ($\widetilde{T}$) | 1.29 $\pm$ 0.01 | 0.86 $\pm$ 0.01
> > __________________
> > EquivCNP ($E$) | 2.03 $\pm$ 0.01 | 1.76 $\pm$ 0.01
> >
> > EquivCNP ($\widetilde{E}$) | 2.05 $\pm$ 0.01 | 1.69 $\pm$ 0.01
> >
> > --------------------
> >
> >
> > Similar to the additional results for the synthetic 1-D regression experiments, these results show that all models achieve higher log-likelihood on the context data, demonstrating that our models are not “underfitting” to the context data.
> >
> > **Equivariance error environmental data**
> >
> > Table 2: Equivariance error on the 2-D environmental regression experiment. Note that all equivariance errors for the US are zero, as the additional fixed inputs are set to 0.
> > _________________
> > Model | Europe
> > _____________________
> > PT-TNP ($\widetilde{T}$) | 0.0406 $\pm$ 0.0005
> >
> > ConvCNP ($\widetilde{T}$) | 0.0237 $\pm$ 0.0004
> >
> > RelaxedConvCNP ($T$) | 0.0239 $\pm$ 0.0004
> >
> > EquivCNP ($\widetilde{E}$) | 0.0242 $\pm$ 0.0005
> > _________________
> >
> > These vary between 2-4%, again indicating that the model’s predictions only deviate slightly from the equivariant predictions.
> >
> > We hope that our previous comment, together with these additional results, have adequately addressed all of your concerns. If so, we would encourage you to reconsider your score.

---

> > ### Author Response · Authors · 2024-08-14
> > **Awaiting your response**
> >
> > Dear dBf7,
> >
> > This is a gentle reminder that we are still awaiting your response. Please note that the discussion period is ending in a few hours.
> >
> > In case it helps, we would like to clarify a technical point that we have left implicit. By showing that the approx. equiv. predictions are approximately equal to equivariant predictions, we show that `d(approx-np(D), equiv-np(D))` is small for all `D` (where `d` is some metric). Then, by the triangle inequality, we find that `d(T approx-np(D), approx-np(T D))` is small for all `T` and `D`:
> >
> > ```
> >     d(T approx-np(D), approx-np(T D))
> >         <= d(T approx-np(D), T equiv-np(D)) + d(T equiv-np(D), equiv-np(T D))
> >             + d(equiv-np(T D)), approx-np(T D))
> > ```
> >
> > where the first and third terms are small because `d(approx-np(D), equiv-np(D))` is small for all `D`, and the second term is zero because `equiv-np` is exactly equivariant. Therefore, the metric which we computed exactly implies equivariance in an approximate sense.

---

> ### Author Response · Authors · 2024-08-12
>
> Thank you for your reply and we hope to address your remaining concerns by providing the metrics you requested.
>
> **Context Log-Likelihood**
>
> We computed the log-likelihoods for reconstructing the context data in the GP experiment, and we will provide the results on the environmental data by the end of the discussion period. Please see the results below, which we will include in the supplementary material.
>
> Table 1. Context log-likelihood for the GP experiment. Ground truth log-likelihood is $0.2806 \pm 0.0005$.
> | TNP                            | TNP ($T$)                            | TNP ($\widetilde{T}$) |
> | ------------------- | ------------------------------ | ------------------------------------ |
> | 0.2296 $\pm$ 0.0007                    | 0.2396 $\pm$ 0.0013                                          | 0.2344 $\pm$ 0.0020                                                      |
> | **ConvCNP ($T$)**                      | **ConvCNP ($\widetilde{T}$)**                                | **RelaxedConvCNP ($\widetilde{T}$)**                                     |
> | 0.2362 $\pm$ 0.0007                    | 0.2218 $\pm$ 0.0009                                          | 0.2381 $\pm$ 0.0008                                                      |
> | **EquivCNP ($E$)**                     | **EquivCNP ($\widetilde{E}$)**                               |                                                                          |
> | 0.2213 $\pm$ 0.0007                    | 0.1992 $\pm$ 0.0010                                          |                                                                          |
> |                                        |                                                              |                                                                          |
>
> Note that these log-likelihoods are much higher than target log-likelihoods reported in the main body, because the models can accurately reconstruct the context data. Moreover, they are close to the ground truth log-likelihood ($0.2806 \pm 0.0005$), indicating that underfitting is unlikely.
>
> **“However, high uncertainty in the reconstruction of context indicates merely inferring the whole sequence from a pattern in the context, which is not a stochastic process. (...) using imputation models over neural processes.”**
>
> A neural process is defined as a neural network architecture that maps a context set to a stochastic process. This stochastic process can then be evaluated at any target inputs to produce predictions for any target set. All models in our submission are of this form. Therefore, by construction, all neural process models derive their predictions from an underlying stochastic process, and this underlying stochastic process can always be queried to e.g. analyse statistics of the underlying time series.
>
> The premise of the neural process framework is that learning to predict the target sets given the context sets makes this underlying learned stochastic process converge to the “true stochastic process”. The sampling distributions of the context and target data should be set up in a way that enables this convergence in the limit of infinite data.
>
> **“Target performance is not easily manipulated by variance, but it can be artificially enhanced by sacrificing context performance, as seen in TNP.”**
>
> If all possible target sets include all possible context sets too, which is usually the case, then we would argue that this is false: in this case, worse context performance directly implies worse target performance, because context sets are target sets. A model could make a trade-off where it specifically performs worse in context set reconstruction in favour of better performance in interpolation further away, but this is unlikely to happen as the objective usually weights all target sets equally. In our experience, we have not seen this happen, unless the neural process severely underfits (which does happen for some architectures, like the original CNP based on just deep sets). In our submission, we do not believe that any of the models severely underfit in any of the experiments, which can be verified by the visualised predictions.

---

> ### Author Response · Authors · 2024-08-12
>
> **Equivariance Error**
>
> While you acknowledge that we (1) show that the approximately equivariant models outperform the equivariant models (with added inputs set to zero) and (2) approximately equivariant model outperform the non-equivariant models too, we understand that your main concerns are that we (a) do not show that our model is equivariant in an approximate sense and (b) do not show that the model’s approximate equivariance reflects the “actual approximate equivariance”. We address these points in order.
>
> (a): Figures 1 and 2 show that the predictive means of the approximately equivariant model are equal to the predictive mean of the corresponding equivariant model plus a “small perturbation”. In other words, visually, the approximate equivariant predictive mean only slightly deviates from the corresponding equivariant predictive mean. Therefore, given that these figures are illustrative of the general behaviour of the models, we believe that we have clearly qualitatively shown that the predictions of the approximately equivariant models are indeed equivariant in an approximate sense.
>
> To also argue this quantitatively, we have computed equiv_deviation = $\mathbb{E}$[norm(equiv_mean - approx_equiv_mean) / norm(equiv_mean)], which quantifies exactly how much on average the predictive means deviate from the corresponding equivariant predictive mean. The results are as follows:
>
> Table 2. Equivariance error for the GP experiment.
> | TNP ($\widetilde{T})$     | ConvCNP ($\widetilde{T}$)        | RelaxedConvCNP ($\widetilde{T}$) | EquivCNP ($\widetilde{T}$) |
> | --------------------- | ------------------------- | -------------------------------- | -------------------------- |
> | 0.0896 $\pm$ 0.0011                        | 0.0823 $\pm$ 0.0005                                | 0.1460 $\pm$ 0.0006                                              | 0.0825 $\pm$ 0.0006                                  |
> |                                            |                                                    |                                                                  |                                                      |
>
>
>
> These percentages are around 8-9% (only the RelaxedConvCNP shows 15% difference, which is based on the approach from Wang et al. [2022a], rather than our approach), which means that, on average, the models’ predictions deviate only slightly from the equivariant predictions.
>
> (b): Whether the models learn the “actual approximate equivariance” is hard to determine. For example, in the GP experiments, what would the actual approximate equivariance be? In addition, what would the actual approximate equivariance be in the climate experiments?
>
> While this is a hard question, we agree that it is an important question, which is why we attempted to answer this question in the following way: the models learn the “actual approximate equivariance” if the perturbation w.r.t. the corresponding equivariant prediction is “consistent with the structure of the problem”. In the GP experiments, we show that the perturbation exactly corresponds with the length-scale change. In the climate experiments, we show that the perturbation exactly corresponds with the key missing information that primarily breaks stationarity of the weather: orography. Therefore, we believe that we have reasonably demonstrated that the models approximate the “actual approximate equivariance”.
>
> By having provided the log-likelihoods for the reconstruction of the context data and having provided numerical evidence that the models' predictions are indeed equivariant in an approximate sense, we hope to have addressed your concerns and consequently hope that you will reconsider your score.

---

### Official Review · Reviewer_WhxG · 2024-07-09

**Soundness:** 3
**Presentation:** 3
**Contribution:** 4
**Rating:** 8
**Confidence:** 5

**Summary:**

The paper describes a new framework for soft/approximate equivariance, based on the functional analysis of compact on hilbert spaces, assuming unitary group actions.

This is then applied to equivariant neural processes.

**Strengths:**

The proposed method is very general and conceptually well grounded, and to the extent I could verify they are also new.

The experiments are well presented and seem promising in terms of accuracy (not sure about efficiency).

Edit after rebuttal: the added experiments are convincing enough in terms of accuracy as well.

**Weaknesses:**

Edit after rebuttal: I think that the method is a good addition to the literature, and my concerns 1-3 below have been adequately answered in the rebuttal and "global rebuttal" parts.

-------------------

The main enigma about this paper, is: does the technique used really warrant putting "neural processes" in the paper's title? It seems like the authors develop a large setup for general approximate equivariance, and the application to neural processes is just one of many applications.

What is restricting the applicability of the method, from general equivariant neural networks, to the class of neural processes? This (meaning, the emphasis on neural processes, and the paper's title) is a big distraction in reading the paper, as one tries to find a reason why the setting is restricted like that and one doesn't actually find a convincing answer.

1) The assumption of having tasks with underlying compact operators seems to be swept under the carpet, and not properly discussed.
In functional analysis "compact" is roughly equivalent to "approximable by finite-rank, as pointed out at several points in the paper, however the authors don't discuss what limitation this could imply. If this is beyond the scope of this work, still I think that it needs highlighting and *at the very least* it requires pointing out very clearly that this is a future direction to be investigated.

2) when authors work with multilinear CNNs (see question 7 below), this seems not well described, and remains probably too mysterious. Also, I think that their implementation becomes clumsy and inefficient: is that so, or do the authors have a justification and complexity control of why not? This is not required to do in much detail, but maybe just to mention that in order not to hide possible underlying difficulties of the method.

3) the parameter $k_n$ from Theorem 3 is not well behaved, so it is not clear if this theorem is useful in practice.

**Questions:**

0) see the un-numbered remark at the beginning of "weaknesses" section, about relevance to "neural processes" to the presented methods.

1) line 49-50 can the authors be more specific about what it means that "any equivariant [...] fixed inputs"? A reference to the actual result would suffice.

2) line 69-70 "represented with stochastic processes" -- is it not "represented by"? (this is a genuine question, not a rethorical one)

3) The statement of theorem 1 is not well written. If $\Phi$ has image in $C(\mathcal X,\Theta)$ and if eq. (2) holds, then $\rho$ has image in the same space, not in $\mathbb R^{2D}$. Similar incompatibilities also hold for the other domain / codomain spaces for operators $\rho, e, \psi$ in that theorem. Also, the discussion in lines 114-116 is incompatible in terms of domain/codomain spaces. This makes the theorem hard or impossible to understand.

4) Still for eq. (2), it is not clear why $\psi$ has two arguments, is it a kernel operator? the notation from (2) should be expanded and actually explained with a degree of consistency/precision.. pretty please..

5) lines 130-133, one can just require that the action of $G$ is by linear unitary operators, which I think is equivalent to what the authors claim? Also, why do you say "acting linearly" at line 130, and then write the additivity explicitly at line 132, as if it were a new requirement and not part of the preceding one?

6) I think that before Proposition 1, or actually even in the introduction itself, one should emphasize the role of compact operators on Hilbert spaces. This is a strong requirement of this theory, and a strong limitation, so it should be highlighted. (I'll add an obvious remark: Don't worry, nobody will consider you worthless if you are fair about the limitations, on the contrary, it speaks highly of you if you do so!)

7) line 156: "$T\simeq CNN(\cdot, t_1,\dots,t_n)$" means what exactly? What is the kind of multilinear CNN's that this refers to? Any reference for them? I think that currently this kind of CNN is not well developed, and it is not that trivial as an extension of usual linear CNNs. I think this should be mentioned. If one looks at appendix C, lines 592 and following, that's just the simplest case of such CNNs, so I maybe the authors could spend some time to point this out, and to describe a little bit the difficulties and issues with such multilinear-CNN theory.

8) in the paragraph lines 205-211 I think that it should be stressed (or at the very least mentioned) that $T$  is assumed to be compact, and can't be more general than that.
9) same mention of "compact" should be inserted also in paragraph lines 249-261
10) same mention of "compact" operators should be inserted in seciton 6 (conclusions section)

**Limitations:**

these are addressed in the "weaknesses" section.

I think that the main limitations are in efficiency and scalability of multi-linear CNNs and in the fact that operators of interest are far from compact and not easy to approximate with finite-range ones.

---

> ### Author Rebuttal · Authors · 2024-08-07
>
> Thank you for acknowledging the novelty of our approach, as well as its theoretical groundness, coupled with encouraging empirical evaluations. We address your concerns below.
>
> **Focus on NPs** - Thank you for highlighting an important point about the general purpose of our approach. The reason why we focussed on the applications in NPs is because we identified a limitation of neural processes that hurts performance in practice. Our goal was to alleviate this limitation and hence developed theory and a method to enable them to model approximate equivariance. We noticed that the theory can be applied more generally and tried to reflect that in the presentation of our paper, but our intended scope is to analyse approximate equivariance in NPs.
>
> One point we would like to make is that NPs are a rather large class of models (and hence, not that restrictive), as many meta-learning methods can be seen as close relatives/variations of NPs.
>
> **Compactness** - Compactness is a notion conventionally only used for linear operators. For nonlinear operators, which is the main focus of the paper, we focus on the (for linear operators equivalent) notion that the inputs and outputs of $T$ can be approximated with a single finite-dimensional basis (Prop. 2 in App A). To see whether this notion is reasonable to assume, suppose e.g. the case where $\mathbb{H} = L^2([a, b])$, which would be appropriate for data living on an interval. Considering the Fourier basis, “compactness” here would mean that, uniformly over the input and output functions of $T$, the magnitude of high-frequency basis elements go to zero as the frequency increases. To us, this is a reasonable assumption.
>
> Our intention was definitely not to sweep this issue under the rug. We will add the above argument to the paragraph in lines 188–194 and emphasise its importance appropriately.
>
> **Multilinear CNNs** - The example with the CNN is to provide intuition for how Theorem 2 could work in the nonlinear case (note that in Theorem 2 $E_n$ is nonlinear, so a nonlinear CNN is suitable in approximating linear non-equivariant operators). We do not intend to propose multilinear CNNs as a practical method—that is, both here and in the experiments, we consider nonlinear CNNs, for which there exist a variety of ways in which the additional fixed inputs can be included into the CNN (e.g. as additional input channels). However, note that multilinear CNN are just one-layer CNNs (without nonlinearities) that take in more than one channel. For example, consider a one-layer CNNs that operates on images with a red, blue, and green channel: CNN(red, blue, green). Extending it with even more channels would give the example in the paper: CNN(red, blue, green, $t_1$, $t_2$, …). We will clarify in the writing that this is only an example to provide intuition. In terms of exposition, Theorem 2 is intended as a step-up to Theorem 3; see the first paragraph of Section 3.
>
> **$k_n$ not well behaved** - The utility of Theorem 3 is to motivate the construction of augmenting with additional, fixed inputs, and we show in the experiments that this construction can yield practical benefits. We agree that Theorem 3 cannot be used to characterise exactly how many additional inputs are needed, because the estimate on $k_n$ is poor. As we mention in App A, better proof techniques and/or additional assumptions may enable better estimates on $k_n$. For example, assuming linearity in Theorem 3 would yield that $k_n$ grows linearly in $n$, which is the result of Theorem 2. We acknowledge this as a limitation of our approach in the Conclusion section, and we are excited to pursue this line of work in order to more accurately identify how the behaviour of $k_n$ depends on $n$ for more general nonlinear operators. We suspect, unfortunately, that such estimates require strong additional assumptions on $T$ and might depend on unknowable constants in practice.
>
> Nevertheless, please refer to the additional experiment that analyses the performance of the models as we increase the number of additional fixed inputs. As shown in the environmental data case (which is a real-life dataset), in this case $k_n$ is not ill-behaved.
>
> **L49-50** - This refers to Theorem 2 and 3, which we will make clear in the revised version of the paper.
>
> **L69-70** - We are happy to rephrase this to “represented by”, to avoid possible confusion as to whether or not predictions are “represented alongside” stochastic processes.
>
> **Theorem 1** - Many thanks for pointing out this mistake. The first $\rho$ should be $\phi: \mathcal{Y} \rightarrow \mathbb{R}^{2D}$, which we will correct in the revised version. Regarding the incompatibilities in the following paragraph, Theorem 1 can be obtained by choosing $\mathcal{Z} = \mathbb{R}^{2D}$. However, in practice we do not enforce this restriction, typically choosing $\mathcal{Z} = \mathbb{R}^E$ where $E >> 2D$.
>
> **$\psi$ in Eq. 2** - Yes, as described on line 111 $\psi$ is a G-invariant positive-definite kernel. We appreciate that the notation used in this theorem is very dense, and we will rephrase to clarify.
>
> **L130-133** - Our intention was to be extra clear. We will slightly reword to avoid the suggestion that it is a new requirement.
>
> **Compact operators** - We did not highlight compactness because it is conventionally a notion for linear operators, whereas our main result is for nonlinear operators (Theorem 3). Nevertheless, instead of compactness, you’re completely right that Theorem 3 requires other technical conditions. We will add a clear mention to these technical conditions in the introduction and discussion.
>
> **L156** - Please see the response above about multilinear CNNs above.
>
> **L205-211, L249-261** - We will clarify in both places. Thank you.
>
> We are very thankful for your detailed feedback, and for the questions / concerns you have raised—we have no doubt that these have made for a stronger paper. We hope that we have adequately addressed your concerns.

---

> > ### Comment · Reviewer_WhxG · 2024-08-08
> > **about the compactness + need more details for question 7**
> >
> > Thank you for the reply, I'm slowly going through the things you wrote back, so this may not be my only comment (sorry).
> >
> > About compactness, if you write something like the reply you put, I'll be satisfied. I was worried only about the "sweeping under the rug" nothing else.
> >
> > One part that i still do not understand is what are multilinear CNNs.
> >
> > a) is there a reference for multilinear CNNs in the literature?
> > b) can you give a formula/pseudocode/details for a special case that is a multilinear CNN but not a standard CNN please? Because I'm not fully sure that I understand.

---

> > > ### Author Response · Authors · 2024-08-08
> > >
> > > We greatly appreciate you taking the time to go through our rebuttal. Please don't hesitate to leave additional comments/questions. We are more than happy to answer them.
> > >
> > > **Compactness**: we will update the paper with the above discussion on compactness.
> > >
> > > **Multilinear CNNs**: With regards to multilinear CNNs: it seems as though there's potentially some communication error here. To be clear, we do not use, nor propose the use of, multilinear CNNs---we use regular (nonlinear) CNNs throughout. Admittedly, we hadn't heard of a "multilinear CNN" before it was mentioned in your review, and were hoping that you would be able to provide a reference.
> > >
> > > Our rebuttal assumed you implied the use of a single-layer CNN without any nonlinearities (which would indeed make it a multiple input CNN with linear operations on the input). The only reference we can find in the literature ([1]) takes this approach.
> > >
> > > [1] Pinson, Hannah, Joeri Lenaerts, and Vincent Ginis. "Linear CNNs discover the statistical structure of the dataset using only the most dominant frequencies." International Conference on Machine Learning. PMLR, 2023.

---

> > > > ### Comment · Reviewer_WhxG · 2024-08-08
> > > >
> > > > Ok sorry, I had forgotten that I was the one to use the term "multilinear" in the first place. I was assuming that the CNN was linear in $t_1,t_2,\dots$, but actually you never said it was.
> > > >
> > > > But anyway, can you give some details on the architecture of the CNN abbreviated at line 156 of the paper?

---

> ### Author Response · Authors · 2024-08-09
>
> No problem at all!
>
> **CNNs**
>
> So, as mentioned, we use a regular CNN which consists of multiple convolutional layers, each of which takes as input a $C_{in}$ dimensional feature map $f: \mathcal{X} \rightarrow \mathbb{R}^{C_{in}}$, where $\mathcal{X}$ denotes a discretised input domain (i.e. grid) and convolves it with a kernel $\psi: \mathcal{X} \rightarrow \mathbb{R}^{C_{out}\times C_{in}}$ : $$(f \ast \psi)(x)= \Sigma_{x'\in \mathcal{X}} f(x')\psi(x - x').$$
>
> here, $C_{in}$ and $C_{out}$ denote the number of input and output channels, respectively, and $\ast$ denotes the convolution operation.  In practice, the kernel has a finite receptive field, meaning that $\psi(x - x') = 0$ when $x - x'$ exceeds some value. After each convolutional layer there is a nonlinearity $\phi: \mathbb{R} \to \mathbb{R}$ which acts point-wise on the feature map value. We have omitted details such as strides, padding and bias, which are also used when implementing. Note that we provide precise details on our hyperparameters choices (e.g. number of layers, number of channels at each layer, receptive field) in the Appendix.
>
> When used in ConvCNPs, the input into the CNN is the output of the 'SetConv' encoder $e: \mathcal{S} \to \mathbb{H}$ which maps datasets $\mathcal{D} \in \mathcal{S}$ to feature maps $f: \mathcal{X} \to \mathbb{R}^{E} \in \mathbb{H}$ .
>
> **Additional fixed inputs**
>
> As we have described, there are many possible ways to include additional fixed inputs $t_i: \mathcal{X} \to \mathbb{R}^{E}$ . The simplest approach is to include them as additional channels into the first convolutional layer, which in practice is achieved by concatenating $f$ with the fixed inputs. The first convolutional layer is then $$(\operatorname{cat}(f, t_1, \ldots, t_B) \ast \psi)(x) = \sum_{x' \in \mathcal{X}} \operatorname{cat}(f, t_1, \ldots, t_B)(x') \psi(x - x').$$ Let $T_{\tau}f = f(\cdot - \tau)$. Note that  $\operatorname{cat}(T_{\tau}f, t_1, \ldots, t_B) \neq T_{\tau}\operatorname{cat}(f, t_1, \ldots, t_B)$ , hence translation equivariance is broken. An alternative approach, which we take in the paper, is to add the additional channels to $f$: $$((f + t_1 + \ldots + t_B) \ast \psi)(x) = \sum_{x' \in \mathcal{X}} (f + t_1 + \ldots + t_B)(x') \psi(x - x').$$ Again, as $T_{\tau}f + t_1 + \ldots + t_B \neq T_{\tau}(f + t_1 + \ldots + t_B)$, translation equivariance is broken.
>
> Remarkably, two of the most prominent approaches to achieving approximate equivariance [1, 2] can be understood as alternative ways of including additional fixed inputs into a CNN to break equivariance. We discuss this in more detail in Appendix C. We believe that this marks an important contribution of our work: to first understand approximate equivariance, to unify existing methods, and to use this to develop a simple and effective approach to building approximate equivariance in *any* operator.
>
> **References**
>
> [1]: Wang, Rui, Robin Walters, and Rose Yu. "Approximately equivariant networks for imperfectly symmetric dynamics." International Conference on Machine Learning. PMLR, 2022.
>
> [2]: van der Ouderaa, Tycho, David W. Romero, and Mark van der Wilk. "Relaxing equivariance constraints with non-stationary continuous filters." Advances in Neural Information Processing Systems 35 (2022): 33818-33830.

---

> > ### Comment · Reviewer_WhxG · 2024-08-09
> >
> > Thanks for the clarification. Given your willingness to clarify the "compactness" part, the above clarifications, and the part of the "global rebuttal" on $k_n$ behavior, that we didn't discuss, I'm more confident to raise my score to 8.

---

> > > ### Author Response · Authors · 2024-08-10
> > >
> > > We greatly appreciate your engagement with our work. Thank you again for taking the time to review the paper in detail—without doubt, your feedback has led to an improvement in the exposition of our research.

---

### Official Review · Reviewer_Ks9f · 2024-07-11

**Soundness:** 3
**Presentation:** 3
**Contribution:** 2
**Rating:** 6
**Confidence:** 2

**Summary:**

This work considers approximately equivariant models --- which may better model or learn real-world tasks than exactly equivariant models --- especially in the context of neural processes. A new approximately equivariant method is developed, which uses an exactly equivariant model along with fixed inputs that break equivariance. This method can be used to parameterize approximately equivariant neural processes, by modifying existing approaches in simple ways. Experiments on 1-D regression, smoke plumes, and environmental data show benefits of the approach.

**Strengths:**

1. Nice flexible framework for approximate equivariance, which unifies existing methods. Similar ideas could be useful more broadly in learning with (approximate) symmetries.
2. The approach has nice ways of controlling the degree of equivariance, by number of inputs ("degrees of freedom" in Section 3.1), or empirically doing things like regularizing the effect of the additional inputs towards zero. The method of setting $t_b(x) = 0$ during test time is also nice.
3. Good empirical gains on several datasets, by making simple tweaks to several NP methods.

**Weaknesses:**

1. Notation and definitions are heavy, which is somewhat understandable given the subject matter, but I do think it can be improved. For instance, compact operators are not defined.
2. From what I can tell, the smoke plumes experiment in Section 5.2 follows a similar setup to that of Wang et al. [2022a], but this is not mentioned. Also it would be good to note how you chose e.g. the parameters of the simulations and the choice to include an obstacle, or whether this was mostly arbitrary (not a big issue).

**Questions:**

1. In equation (3), I think think the number of inputs should be larger than $n$, right?
2. Perhaps I missed this, but do you initialize the models to be equivariant? Via setting $t$ to be the zero function?
3. It would be worth noting connections of your approximate equivariance approach to some related prior work. You mention the connection to positional embeddings in Transformers on Page 5, but this also applies to other domains such as Vision Transformers [Dosovitskiy et al. 2021]; this is more specifically investigated in the context of approximate equivariance by Lim et al. 2023. Also, the approach of taking additional inputs is related to the symmetry breaking approach of Xie et al. 2024 and the probabilistically / approximately equivariant approach of Kim et al. 2023.

References:
* Wang et al. 2022a. Approximately Equivariant Networks for Imperfectly Symmetric Dynamics
* Dosovitskiy et al. 2021. An Image is Worth 16x16 Words: Transformers for Image Recognition at Scale
* Lim et al. 2023. Positional Encodings as Group Representations: A Unified Framework
* Xie et al. 2024. Equivariant Symmetry Breaking Sets
* Kim et al. 2023. Learning Probabilistic Symmetrization for Architecture Agnostic Equivariance

**Limitations:**

Some discussion in conclusion

---

> ### Author Rebuttal · Authors · 2024-08-07
>
> Thank you for acknowledging our theoretical and empirical contributions in the field of approximate equivariance, with a focus on applications in neural processes. In particular, we are pleased that you appreciate the importance of setting the additional inputs to zero at test time outside the training domain, which we believe is one of the key components for the model’s ability to generalise. We address your concerns below.
>
> **Weaknesses**
>
> **Heavy notation** - We acknowledge that the paper might be notation and definition heavy at times given the nature of the subject we tackle, but we put in effort towards including as many intuitive explanations as possible—for example, the discussion after Theorem 2 (lines 154 - 171).
>
> If you have further suggestions to improve notational clarity, we would be open to them.
>
> **Definition of compactness** -  We apologise for not including a definition of compactness and we will update the text accordingly.
>
> **Smoke plume setup** - You are right, the setup does hold similarities to that of Wang et al. [2022a], but there are also a few key differences, especially in the way we break the strict equivariance. These are all outlined in Appendix D.2, but here we provide a comparison with the setup from Wang et al. [2022a].
>
> **Similarities**
>
> - We also use the PhiFlow library in order to generate the smoke simulations.
>
> - We use the same solver techniques (i.e., the code for the solver is inspired from the repository associated with Wang et al. [2022a]), with the same parameters for the time discretisation and choice of optimiser.
>
> **Differences**
>
> - Wang et al. [2022a] sample the smoke inflow location out of $35$ locations, we randomly sample out of $3$ locations $((30, 5), (64, 5), (110, 5)$ on a ($128, 128$) domain).
>
> - We randomly sample the buoyancy force at each simulation as $B \sim \mathcal{U}[0.1, 0.5]$, and keep it constant throughout the entire sample domain. In contrast, Wang et al. [2022a] have two subdomains, with a different buoyancy force for each subdomain, and they keep those values constant for all their simulations.
>
> - Besides the boundary of the simulation (which are the same in our and Wang et al [2022a]’s setup), we also consider an obstacle with its centre at $(64, 100)$ and a radius of $20$.
>
> - We train and evaluate the model on randomly sampled $32 \times 32$ patches of the $128 \times 128$ domain corresponding to the final simulation state, as opposed to the entire state ($128 \times 128$).
>
> Thus, although we took inspiration from Wang et al. [2022a] setup, there are some key differences—we break the symmetry of the system through the closed boundary (as in Wang et. al [2022a]), through the fixed spherical obstacle through which smoke cannot pass, and by sampling the inflow position. The latter is also performed by Wang et. al [2022a], but they consider more possible inflow positions. In the limit of randomly sampling the inflow position, this no longer breaks the symmetry of the system, so we decided to only use 3 positions in order to depart from strict symmetry more.
>
> **Questions**
>
> **Number of inputs** - Many thanks for pointing out this mistake—the number of inputs should be $2n + 1$, which we will correct in the revised version of the paper.
>
> **Model initialisation** - We do not initialise the models to be equivariant, but there are two techniques that we use in order to recover equivariance out-of-distribution, where the training data does not contain enough information in order for the model to correctly learn how to depart from strict equivariance. More specifically, we:
> - During training, randomly set the additional inputs to $0$ (corresponding to strict equivariance) with a fixed probability (usually $0.5$), such that the model learns a good strict equivariant representation of the underlying system when the additional inputs are zeroed out.
>
> - During testing, we always set the additional inputs outside the training domain to $0$, such that the model reverts to the predictions of the equivariant component in the regions where it has not seen any data during training (where strict equivariance is the most reasonable inductive bias we can assume).
>
> See the paragraph “Recovering equivariance out-of-distribution” at line 224.
>
> **Connections to related work** - You are correct that this work has connections to, and offers insights for, many other important methods used within ML such as positional encodings in ViT. Many thanks for pointing out Lim et al. 2023, Xie et al. 2024 and Kim et al. 2023. These three works are indeed related to our work, and we will include them as citations in the related work section.
>
> Thanks again for your feedback—we hope that we have adequately addressed your concerns. If so, we would be grateful if you could consider increasing your score.

---

> > ### Comment · Reviewer_Ks9f · 2024-08-08
> >
> > We thank the authors for their rebuttal. The clarifications are useful for me.
> >
> > On the Wang et al. [2022a] point, thanks for clarifying the differences. My point is that, given that you take inspiration from and use the same tools as the Wang et al. setup, you should note and cite this in the paper.
> >
> > Everything else looks good to me, I maintain my score.

---

> > > ### Author Response · Authors · 2024-08-09
> > >
> > > Many thanks for your reply. You raise an important point that Wang et al. should be cited when detailing the set up for this experiment---we shall modify the paper accordingly.
> > >
> > > Thank you again for your feedback, and please don't hesitate to ask any remaining questions should they arise.

---

### Official Review · Reviewer_7yUH · 2024-07-14

**Soundness:** 3
**Presentation:** 2
**Contribution:** 2
**Rating:** 5
**Confidence:** 3

**Summary:**

The work provides an alternative approach to obtain a loose equivariance constraint. The authors established an interesting relationship between equivariant and non-equivariant models, showing that equivariant models with enough fixed input can approximate any non-equivariant model. Subsequently, the authors suggest using additional fixed input in neural processes, which is a common practice in transformers and NeRFs. The proposed method achieved better performance on multiple benchmark datasets.

**Strengths:**

1. The work provided a nice theoretical result that offers an interesting reinterpretation of existing practices.

2. The proposed technique is empirically evaluated on a wide range of datasets.

**Weaknesses:**

The work provides a nicer reinterpretation of the existing practice of adding fixed/learnable positional embedding. Despite this reinterpretation, the work does not quantify the degree to which these additional fixed inputs hurt the equivariance property of the equivariant models.

Nor does it provide a scheme to automatically learn the number of fixed inputs for controlling the equivariance restriction.

Thus, the work indeed proposes an approximately equivariant model, however, with unknown approximation.

This severely limits the contribution of the work.

Line 91: different notation should be used to denote the group action on $X$ and $Y$

Line 102: the symbol $f$ is already used in line 87 to denote the group element. The use of different symbols will fascinate reading.

Line 115: the notation of the encoder matches with a notation of the identity element of the group.

**Questions:**

Line 103: how is the action of the group on the dataset $D$ defined?

Equation 2, line 111: How is the G-invariance of the kernel defined?

Line 143: $E_n$ is defined to take $n$ additional input; however, line 146 is defined to map on $H^{1+2n}$, which I think means $2n$ additional input. An explanation would be appreciated.

**Limitations:**

yes

---

> ### Author Rebuttal · Authors · 2024-08-07
>
> Thank you for acknowledging our theoretical contributions, as well as the empirical evaluation on a variety of datasets. We hope that our work justifies the advantages of relaxing model symmetries and proves that the proposed framework is an effective approach for achieving this. We address your concerns below.
>
> **Weaknesses**
>
> **Quantifying degree of non-equivariance** We believe that our experiments carefully analyse the effects of including additional inputs in equivariant architectures through both quantitative results and visualisations (i.e. we run both the versions with and without the additional fixed inputs, so the results quantify the effect of including additional fixed inputs). We provide results for both the synthetic GP and environmental data experiments, showing how our models break equivariance and how they compare to their strictly equivariant counterparts.
>
> Nevertheless, we acknowledge that we do not characterise precisely how and to which degree additional inputs break equivariance. (Throughout the exposition, however, we do repeat the quite rudimentary intuition that more additional inputs should break equivariance more, e.g. lines 163-171). In the conclusion, we mention this limitation, which paves the way for interesting future work.
>
> **Automatically learn the number of fixed inputs** - We respectfully disagree that not having a way to automatically learn the number of fixed inputs is a severe limitation of the work. In practice, the number of additional inputs can very reasonably be tuned as a hyperparameter alongside all other architectural choices, like the number of layers and widths of MLPs.
>
> In our experiments, setting the number of fixed inputs to a “reasonably high number” tends to work well. We suspect that the dropout procedure (“Recovering equivariance out-of-distribution” on line 224) regularises any unnecessary additional inputs. The additional ablation provided in the common response supports this hypothesis.
>
> **L91** - We acknowledge that using different symbols for the group actions on $\mathcal{X}$ and $\mathcal{Y}$ would further disambiguate the notation. However, the notation in our submission is already heavy, as pointed out by other reviewers, and we remark that overloading $g$ in this way is both accurate (by defining $\mathcal{X}$ and $\mathcal{Y}$ both as $G$-spaces) and standard. For example, see the section “Formalization” of the entry “Equivariant map” on Wikipedia.
>
> **L102** - Many thanks for pointing this out—we will modify the notation such that the symbol $g’$ is used to denote an alternative group element.
>
> **L115** - Thanks for pointing this out, we will modify the notation.
>
> **Questions**
>
> **L103** - We apologise for the confusion, as there is a typo on line 102, where we explain how the action of the group on the dataset $\mathcal{D}$ is defined. $g\mathcal{D} \in S$ consists of input-output pairs ($g\mathbf{x}_n, \mathbf{y}_n$), rather than  pairs ($g\mathbf{x}_n, \mathbf{x}_y$).
>
> **L111** - G-invariance of the kernel $\psi$ is defined by the property that $\psi(\mathbf{x}_n, \mathbf{x}_m) = \psi(g\mathbf{x}_n, g\mathbf{x}_m)$ for all $g \in G$. We will add this definition to the writing.
>
> **L143/L146** - We do apologise for the confusion here: there are indeed $2n$ additional inputs, owing to $n$ $t$’s and $n$ $e$’s. We thank the reviewer for spotting the mistake, and will correct it in the revised version.
>
> In light of the above, we would be grateful if you would reconsider your score.

---

> > ### Comment · Reviewer_7yUH · 2024-08-12
> > **Response to the Authors**
> >
> > Thanks for the response. I want to maintain my score.

---

### Author Rebuttal · Authors · 2024-08-07

We would like to thank all reviewers for the time taken to review the paper, for their feedback and useful suggestions, and we hope to address all their concerns through our responses. We are pleased that, in general, our method was seen as a general way of achieving approximate equivariance, while also being “conceptually well grounded”, “new”, and a “flexible framework” that “unifies existing methods”. Moreover, we are content that the reviewers also appreciated the effectiveness of our method, noting the “good empirical gains on several datasets”.

However, there were some concerns that we aim to address through this general comment, as well as individually with each reviewer. A weakness mentioned by all reviewers is that we do not characterise precisely how many additional input channels are needed and how their number influences the degree of approximate equivariance. We acknowledge this weakness, which is a hard theoretical research question that we aim to investigate in future work. Our suspicion is that no practical answer exists, just like there is no practical way to know how wide an MLP should be for a particular data problem. We, however, do not think that this diminishes the utility of the proposed approach, because the number of additional input channels can be tuned as a hyperparameter. In practice, choosing a “sufficiently high number” tends to just work fine, which is what we did in our empirical evaluations.

Moreover, we have run an ablation study for the 2D environmental regression experiment where we vary the number of additional fixed inputs into the ConvCNP ($\widetilde{T}$) model. For this particular experiment, the information that is primarily missing is topography. Since topography is only a single variable, we expect that a low number of additional inputs will saturate performance. The predicted log-likelihoods for Europe are as follows:

No. fixed inputs: 0 / 1 / 2 / 4 / 8 / 16

Log-likelihood: 1.11 / 1.19 / 1.19 / 1.20 / 1.20 / 1.20

We observe that the largest gain is observed when going for zero to a single fixed input, and that there are diminishing improvements thereafter. Importantly, observe that increasing the number of additional inputs never hurts performance. This demonstrates that the model is able to ignore additional inputs it doesn’t need. It also suggests that one can usually just set the number of additional inputs to a “sufficiently high number”. We suspect that this is due to dropout being included in the training procedure, so that the closest equivariant component is always learnt. We are currently running the same ablation for the synthetic GP experiment, and will provide these results in due course.

These results can also be visualised in Table 1 in the attached PDF.

---

### Decision · Program_Chairs · 2024-09-25

**Decision:**

Accept (poster)

**Comment:**

The paper presents a general formalism for the construction of approximately equivariant operators agnostic to the choice of symmetry group and model architecture, which is then used to construct approximately equivariant neural processes. The method essentially uses an exactly equivariant model together with fixed inputs that break equivariance. The reviewers generally had a positive appraisal of the work, which I mostly share--I think the paper is a good contribution to the literature. However, the paper should be further revised for the camera ready. Some points are highlighted below.

First off, the authors should consider incorporating the various clarifications that they have made during the discussion phase, making use of the extra page. Incorporating these points could amount to a moderately large revision. Beyond the broad discussion points, generally, the discussion to related work in the paper can also be improved. While the paper cites some approximately equivariant networks-based works in the first page, the paper can sometimes read like these works were an afterthought. The idea of approximate equivariance is now embedded in the literature for a few years and this should be prominently reflected in the narrative--this does not come through in reading the abstract, for example. The paper will also benefit in including additional references suggested by reviewer Ks9f. In the same vein, another work about generalization and approximation error characterization that is more relevant to the paper than some papers cited is: "Approximation-generalization trade-offs under (approximate) group equivariance", M Petrache, S Trivedi, NeurIPS 2023, as it is not specific to NNs. Reviewers 7yUH and dBf7 have also raised a valid concern about equivariance error quantification. While it is understandable that a detailed analysis is out of scope for the paper, this limitation should be highlighted more clearly in the paper. Finally, the concerns highlighted by dBf7 about the log-likelihood of context reconstruction should be taken note of. The authors have provided a response in the rebuttal. It would be beneficial to include these results in the paper with some accompanying discussion.